**eLife** RESEARCH ARTICLE

# DePARylation is critical for S phase progression and cell survival

Litong Nie[1†], Chao Wang[1†], Min Huang[1†], Xiaoguang Liu[1], Xu Feng[1], Mengfan Tang[1], Siting Li[1], Qinglei Hang[1], Hongqi Teng[1], Xi Shen[1,2], Li Ma[1], Boyi Gan[1], Junjie Chen[1]*

[1]Department of Experimental Radiation Oncology, The University of Texas MD Anderson Cancer Center, Houston, United States; [2]Institute for Personalized Cancer Therapy, The University of Texas MD Anderson Cancer Center, Houston, United States

*For correspondence:
jchen8@mdanderson.org

†These authors contributed equally to this work

Competing interest: The authors declare that no competing interests exist.

## Abstract

Poly(ADP-ribose)ylation or PARylation by PAR polymerase 1 (PARP1) and dePARylation by poly(ADP-ribose) glycohydrolase (PARG) are equally important for the dynamic regulation of DNA damage response. PARG, the most active dePARylation enzyme, is recruited to sites of DNA damage via pADPr-dependent and PCNA-dependent mechanisms. Targeting dePARylation is considered an alternative strategy to overcome PARP inhibitor resistance. However, precisely how dePARylation functions in normal unperturbed cells remains elusive. To address this challenge, we conducted multiple CRISPR screens and revealed that dePARylation of S phase pADPr by PARG is essential for cell viability. Loss of dePARylation activity initially induced S-phase-specific pADPr signaling, which resulted from unligated Okazaki fragments and eventually led to uncontrolled pADPr accumulation and PARP1/2-dependent cytotoxicity. Moreover, we demonstrated that proteins involved in Okazaki fragment ligation and/or base excision repair regulate pADPr signaling and cell death induced by PARG inhibition. In addition, we determined that PARG expression is critical for cellular sensitivity to PARG inhibition. Additionally, we revealed that PARG is essential for cell survival by suppressing pADPr. Collectively, our data not only identify an essential role for PARG in normal proliferating cells but also provide a potential biomarker for the further development of PARG inhibitors in cancer therapy.

## eLife assessment

The demonstration that the PARG dePARylation enzyme is required in S phase to remove polyADP-ribose (PAR) protein adducts that are generated in response to the presence of unligated Okazaki fragments is potentially **valuable**, but the evidence is **incomplete**, and identification of relevant PARylated PARG substrates in S-phase is needed to understand the role of PARP1-mediated PARylation and PARG-catalyzed dePARylation in S-phase progression.

## Introduction

Poly(ADP-ribosyl)ation or PARylation is a conserved post-translational modification that is important for many cellular processes, including DNA damage repair (*Perina et al., 2014*; *Gupte et al., 2017*). PARylation is mainly driven by poly(ADP-ribose) polymerase 1 (PARP1), and to a lesser extent by PARP2, using NAD + as an ADP-ribose donor and generating nicotinamide (NAM) as a byproduct in human cells. Extensive studies have already established the key roles of PARP1/2 in DNA damage response (DDR; *Liu et al., 2017*; *Azarm and Smith, 2020*). As a DNA damage sensor, PARP1 rapidly recognizes and binds to DNA damage sites, which dramatically activates its own enzymatic activity

and thereby modifies itself and other proteins with pADPr and mono-(ADP-ribosylation). These modifications lead to the recruitment of proteins that are involved in DDR and DNA repair. The binding to DNA by activated PARP1 acts as a 'hit and run' mechanism (*Thomas et al., 2019*), which restricts PARylation at a specific point and facilitates the subsequent engagement of other repair proteins. Thus, PARylation by PARP1, when responding to DNA damage, is a rapid and transient process. In addition, PARP1 can be activated by unligated Okazaki fragments to induce endogenous S phase pADPr, which then recruits XRCC1 and LIG3 to facilitate the ligation of these Okazaki fragments (*Hanzlikova et al., 2018*).

PARylation is a reversible post-translational modification. The removal of pADPr is mainly carried out by poly(ADP-ribose) glycohydrolase (PARG) (*Davidovic et al., 2001*). However, PARG cannot remove the terminal ADP-ribose, whose removal requires additional hydrolases, including terminal ADP-ribose glycohydrolase (TARG1), ADP-ribose-acceptor hydrolases ARH1/3, and possibly other macrodomain-containing proteins, such as hMacroD1/D2 (*Rack et al., 2020*). Later studies showed that PARG and ARH3 (ADPRHL2/ADPRS) are the primary dePARylation enzymes in vertebrates, although ARH3 has much lower activity against pADPr than PARG and mainly functions as a serine-directed mono-ADP-ribosylhydrolase (*Fontana et al., 2017*; *Prokhorova et al., 2021*; *Oka et al., 2006*). As an endo-glycohydrolase and exo-glycohydrolase, PARG specifically hydrolyzes the glycosidic bonds. PARG may function as an oncogene, as the high level of PARG promotes cell transformation and invasion and is associated with poor overall survival (*Marques et al., 2019*). Moreover, complete loss of PARG leads to embryonic lethality in mice (*Koh et al., 2004*). Importantly, PARG is recruited to sites of DNA damage by PARP1- or pADPr -dependent and PCNA-dependent mechanisms (*Mortusewicz et al., 2011*) and involved in both DNA double-strand breaks (DSBs) and single-strand breaks (SSBs) repair. Loss of PARG sensitizes cells to DNA damage agents (*Shirai et al., 2013b*; *Shirai et al., 2013a*; *Fujihara et al., 2009*). In addition, PARG is also involved in transactions at replication forks (*Fathers et al., 2012*; *Margalef et al., 2018*).

On average, a single cell suffers thousands of DNA damage events every day *Ames et al., 1993*; PARylation and dePARylation occur rapidly and collaboratively to facilitate DNA repair in the cell. This rapid PARylation and dePARylation cycle is likely important for homeostasis regulation. Unrestrained PARylation following DNA damage may lead to a significant change in the NAD+ level in cells, which not only inhibits other cellular processes that require NAD+ but also results in cell death due to NAD+ depletion and/or PARP1-dependent cell death called Parthanatos (*Prokhorova et al., 2021*; *Park et al., 2020*). Thus, PARG and other dePARylation enzymes are critically important for the recycling of NAD+ and the control of NAD+ homeostasis (*Mashimo et al., 2013*; *Nagashima et al., 2020*; *Li et al., 2021*).

Targeting PARylation and dePARylation is a promising strategy in cancer and other therapies. PARP inhibitors (PARPis) have already been approved for clinical treatment of cancers with homologous recombination (HR) deficiency, while targeting dePARylation is an encouraging alternative strategy to overcome PARPi resistance (*Gravells et al., 2017*; *Chen and Yu, 2019*; *Pillay et al., 2021*; *Slade, 2020*; *Min and Wang, 2009*). Given the critical roles of PARG in the maintenance of the PARylation/dePARylation cycle in cells, PARG inhibitors have been developed recently as potential anti-cancer agents (*Slade, 2020*). These include PDD00017273 (PARGi; *Figure 1—figure supplement 1A*; *James et al., 2016*), COH34 (*Chen and Yu, 2019*), and JA2131 (*Houl et al., 2019*). Interestingly, cells with deficiency in HR and other DNA damage signaling/repair pathways showed increased sensitivity to the PARG inhibitors PDD00017273 and COH34 (*Gravells et al., 2017*; *Chen and Yu, 2019*). These PARG inhibitors could also sensitize cells to radiation therapy and chemotherapy (*Nagashima et al., 2020*; *Chen and Yu, 2019*; *Houl et al., 2019*; *Gravells et al., 2018*). Thus, there is significant interest in further developing these PARG inhibitors for cancer treatment.

Despite the crucial role of PARG in dePARylation in multiple pathways, detailed knowledge of its mechanism of action remains elusive. Especially, how PARG regulates dePARylation in normal unperturbed cells and the balance between PARylation and dePARylation remain unclear. To address this issue, we performed multiple CRISPR screens and created PARP1/2 DKO, PARG KO, and additional KO cells. We showed that loss of dePARylation activity induced S-phase-specific pADPr signaling, which likely originated from unligated Okazaki fragments. Consequently, the failure to remove S phase pADPr led to PARylation- or PARP1/2-dependent cell death. Furthermore, perturbation of Okazaki fragment ligation and/or base excision repair (BER) increased pADPr signaling and promoted

cell death induced by PARGi. In addition, we revealed that the PARG level is a potential biomarker for PARGi-based cancer therapy. Moreover, we showed that PARG is essential for cell survival, which can be exploited for cancer treatment.

## Results

### PARG depletion leads to drastic sensitivity to PARGi

The results of early studies indicated that defective DNA repair pathways and DNA replication stress would result in enhanced sensitivity to PARG inhibitor, PDD00017273 (PARGi) (*Gravells et al., 2017*; *Chen and Yu, 2019*; *Pillay et al., 2021*; *Ali et al., 2021*; *Pillay et al., 2019*). To further define the key DDR pathways that are important for cellular response to PARGi, we performed CRISPR screening using our homemade DDR sgRNA library, which targets approximately 360 genes involved in various DDR pathways (*Su et al., 2020*). Interestingly, our screen results showed that PARG depletion led to significantly increased cellular sensitivity to PARGi (*Figure 1A*, *Supplementary file 1a*). In addition, we found that POLB (*Figure 1A*), which is important for BER, a repair pathway that relies on PARP1 functions and pADPr-dependent DNA damage signaling, also showed synthetic lethality with PARGi. The synthetic lethality between POLB and PARGi was consistent with the findings of a previous report (*Ali et al., 2021*), indicating the high quality of our screening results.

To further validate our data, we generated PARG KO cells in the same 293 A cells used in DDR sgRNA library screening. Interestingly, PARG KO cells showed extreme sensitivity to PARGi, with nearly thousand-fold sensitivity (the $IC_{50}$ in 293 A cells was 96±24 µM and 210±30 nM in PARG KO cells, on the basis of five independent biological replicates; *Figure 1B*). In addition, we used another two structurally similar compounds (PDD00017272 and PDD00017238) to further confirm the extreme sensitivity of PARG KO cells to PARG inhibition (*Figure 1—figure supplement 1B*), while the inactive but structurally similar compound PDD0031705 did not distinguish wild-type (WT) or PARG KO cells (*Figure 1—figure supplement 1C*). To further validate that this sensitivity was not restricted to one cell line, we generated HeLa-derived PARG KO cells with a different gRNA. As expected, HeLa-derived PARG KO cells also displayed extreme sensitivity to PARGi (*Figure 1—figure supplement 1D*). These results suggest that PARG depletion resulted in drastic sensitivity to PARGi.

We reconstituted PARG KO cells with WT PARG or a PARG catalytic-inactive mutant E755/756 A (*Patel et al., 2005*) to further investigate the nature of PARGi sensitivity in PARG KO cells. Cells treated with the alkylating agent methyl methanesulfonate (MMS), which is known to activate pADPr signaling, were used to test PARG activity. As shown in *Figure 1—figure supplement 1E*, PARG KO cells and cells reconstituted with PARG catalytically inactive mutant showed a significant increase in pADPr signaling induced by MMS, while WT cells or cells reconstituted with WT PARG did not. Expectedly, reconstitution with WT PARG reversed the sensitivity of PARG KO cells to PARGi, while reconstitution with PARG catalytic inactive mutant failed to do so (*Figure 1C*). Together, these data strongly suggest that loss of PARG activity induces dramatic sensitivity to PARGi.

The question is why PARG KO cells would be sensitive to PARGi. There are at least two potential explanations for this observation. First, PARGi may target not only PARG but also a second target, which only becomes essential in the absence of PARG. Second, our PARG KO cells, despite being generated with two different sgRNAs and in two independent cell lines validated by western blotting and DNA sequencing, are not complete PARG KO cells.

There are five reviewed or potential PARG isoforms identified in the Uniprot database. The two different sgRNAs used here target all three catalytically active isoforms (isoforms 1, 2, and 3), and sgRNA#2 used in HeLa cells also targets isoforms 4 and 5, but these two isoforms are considered catalytically inactive according to the Uniprot database (*Figure 1—figure supplement 1D*). Nevertheless, it is possible that sgRNA-mediated genome editing may lead to the creation of new alternatively spliced PARG mRNAs and/or the use of alternative ATG as start codon, which can produce residual but catalytically active forms of PARG. It would be challenging to search for these putative PARG isoforms due to limited PARG antibodies and conceivable low expression of these isoforms. To address this important question, we performed several experiments to test both hypotheses. Please see below for additional studies.

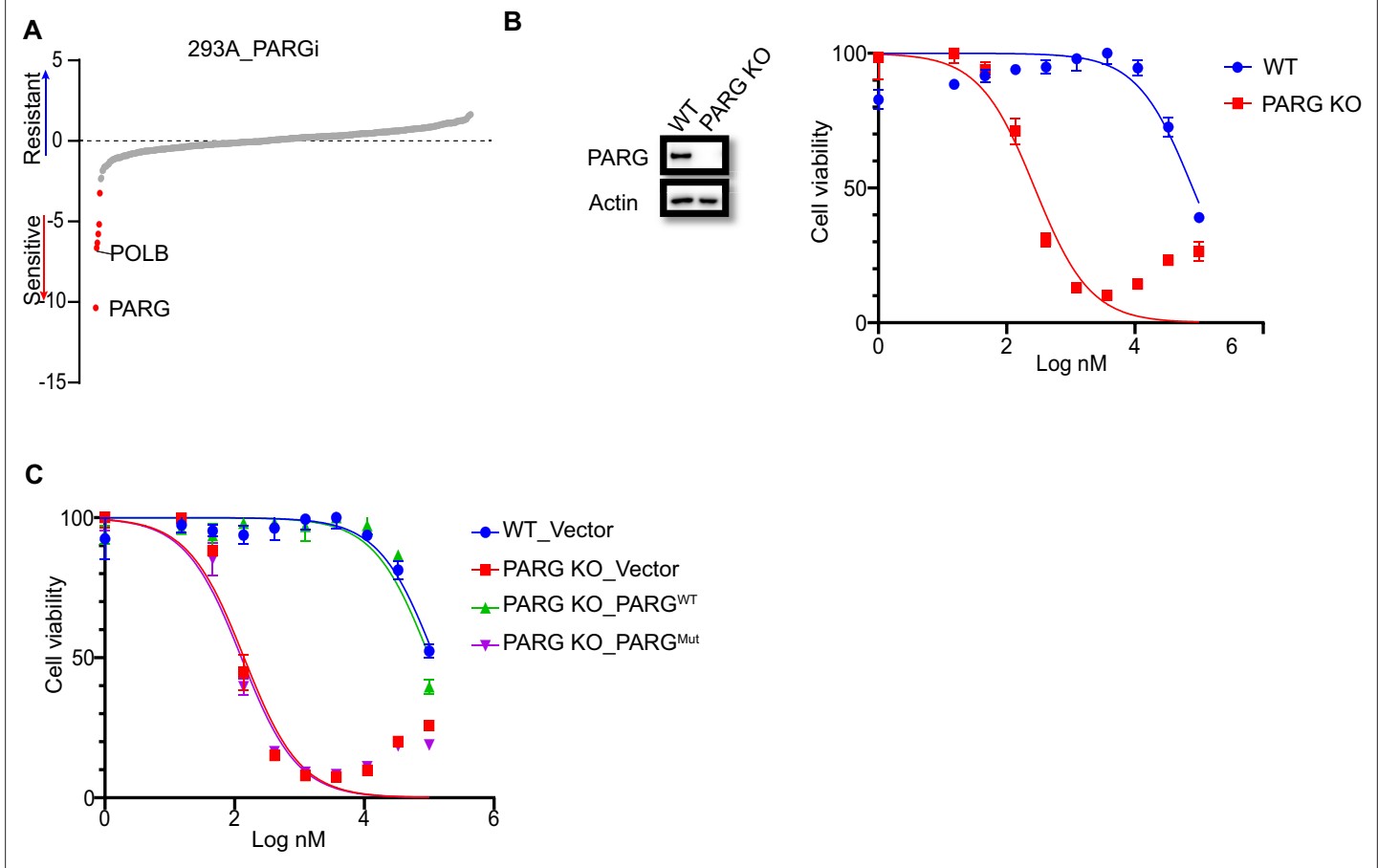

**Figure 1.** PARG loss sensitizes cells to PARGi. (**a**) Ranking of co-essential genes with PARGi treatment on the basis of a DrugZ analysis of the results of CRISPR/Cas9 screening with a DDR library in HEK293A cells. The NormZ score was used to determine a possible synthetic lethality gene under PARGi treatment. Drug-sensitive genes were marked in red; drug-resistant genes were marked in blue on the basis of the false discovery rate (FDR, 0.05 cut-off). (**b**) HEK293A WT cells and HEK293A PARG KO cells were treated with different doses of PARGi for 72 hr. Cell viability was determined by the CellTiter-Glo assay. (**c**) HEK293A PARG KO cells, re-constituted with either full-length PARG or catalytic domain mutation of PARG, were treated with different doses of PARGi for 72 hr. Cell viability was determined by the CellTiter-Glo assay.

The online version of this article includes the following source data and figure supplement(s) for figure 1:

**Source data 1.** DrugZ analysis results of CRISPR/Cas9 screening with DDR library in PARGi treated HEK293A cells.

**Source data 2.** Original file for the western blot analysis in *Figure 1B* and *Figure 1—figure supplement 1D*.

**Source data 3.** PDF containing *Figure 1C*, *Figure 1—figure supplement 1D* and original scans of the relevant western blot analysis (anti-actin and anti-PARG) with highlighted bands and sample labels.

**Figure supplement 1.** PARG KO cells are sensitive to active PARG inhibitors.

**Figure supplement 1—source data 1.** Original file for the western blot analysis in *Figure 1—figure supplement 1E*.

**Figure supplement 1—source data 2.** PDF containing *Figure 1—figure supplement 1E* and original scans of the relevant western blot analysis (anti-actin, anti-pADPr and anti-PARG) with highlighted bands and sample labels.

## PARGi induces PARP1/2-dependent cell death in PARG KO cells

As shown above (*Figure 1—figure supplement 1E*), MMS treatment significantly increased pADPr signaling in PARG KO cells. Interestingly, pADPr signaling further increased modestly in PARG KO cells treated with the combination of PARGi and MMS (*Figure 2—figure supplement 1A*). To further evaluate the underlying mechanisms of PARGi sensitivity in PARG KO cells, we generated PARP1/2 DKO cells in both WT and PARG KO cells and treated them with PARGi (*Figure 2A*). PARP1 cleavage was detected in PARG KO cells treated with PARGi, indicating apoptosis. Moreover, pADPr accumulated significantly in PARG KO cells treated with PARGi, but not in other cells (*Figure 2—figure supplement 1B*). Furthermore, we used flow cytometry to detect pADPr signaling following short

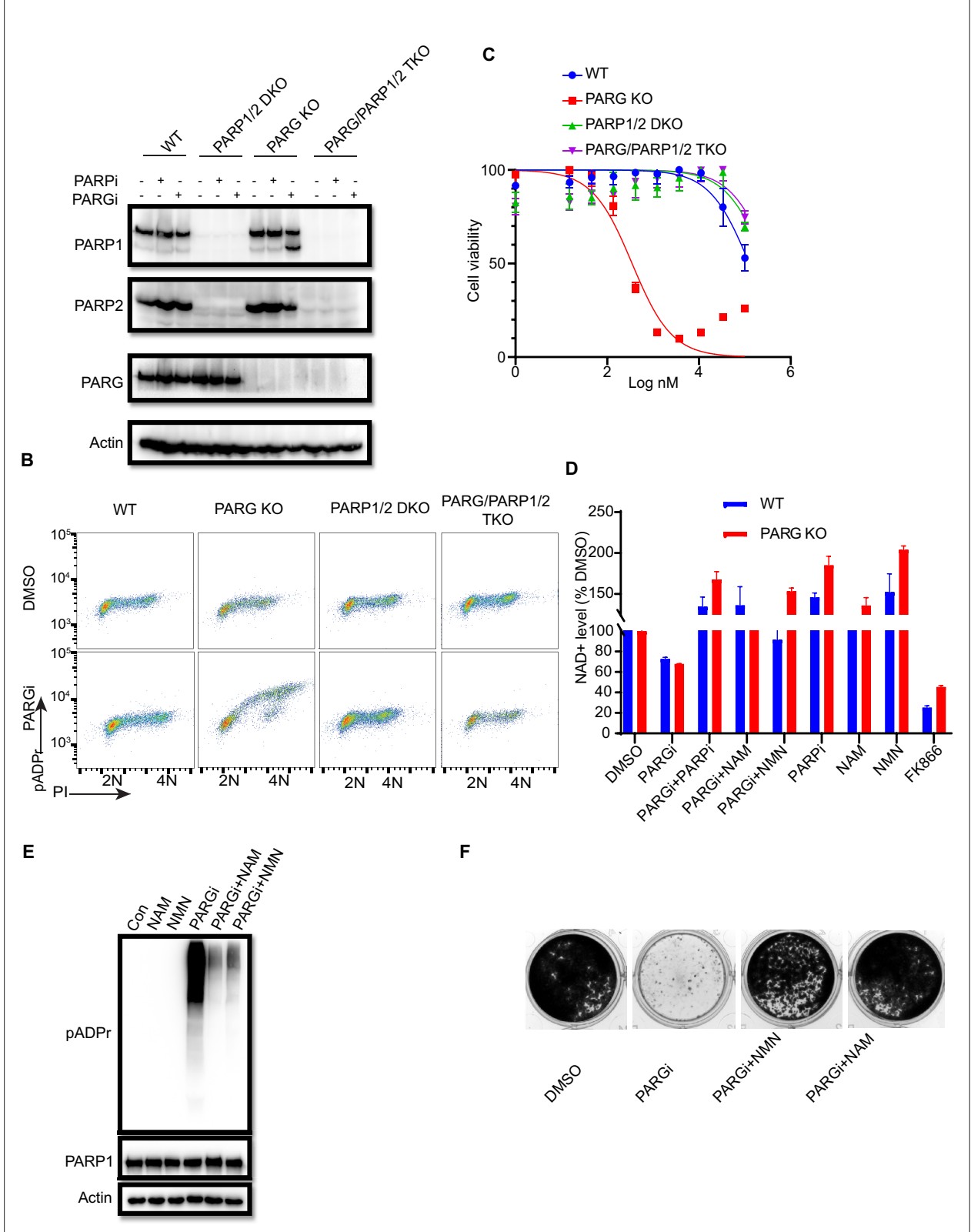

**Figure 2.** PARGi treatment induces NAD+- and PARP-dependent cell death in PARG KO cells. (**a**) HEK293A WT, PARG KO, PARP1/2 DKO, and PARG/PARP1/2 TKO cells were treated with PARGi (1 µM) for 72 hr. The total cell lysates were immunoblotted with the indicated antibodies. (**b**) HEK293A WT, PARG KO, PARP1/2 DKO, and PARG/PARP1/2 TKO cells were treated with DMSO or 10 µM PARGi for 4 hr and then fixed and stained with anti-pADPr antibody and propidium iodide (PI). (**c**) HEK293A WT, PARG KO, PARP1/2 DKO clls, and PARG/PARP1/2 TKO cells were treated with different doses of

*Figure 2 continued on next page*

*Figure 2 continued*

PARGi for 72 hr. Cell viability was determined by the CellTiter-Glo assay. (**d**) Relative NAD $^+$ level in HEK293A WT and PARG KO cells with the indicated treatment for 48 hr. PARGi, 10 μM; PARPi, 10 μM; NAM, 100 μM; NMN, 1 mM; FK866, 10 nM. (**e**) PARG KO cells were treated with PARGi (10 μM) or PARGi and NAM (100 μM) or NMN (1 mM) for 48 hr. The total cell lysates were immunoblotted with the indicated antibodies. (**f**) Results of clonogenic assays conducted using HEK293A PARG KO cells treated with PARGi (500 nM) or PARGi and NAM (100 μM) or NMN (1 mM) for 7 days.

The online version of this article includes the following source data and figure supplement(s) for figure 2:

**Source data 1.** Original file for the western blot analysis and colony formation assay in *Figure 2* and *Figure 2—figure supplement 1B*.

**Source data 2.** PDF containing *Figure 2* and *Figure 2—figure supplement 1B* and original scans of the relevant western blot analysis and colony formation assay with highlighted bands and sample labels.

**Figure supplement 1.** PARP-dependent pADPr and reduced NAD $^+$ may contribute to cell death induced by PARGi treatment in PARG KO cells.

**Figure supplement 1—source data 1.** Original file for the western blot in *Figure 2—figure supplement 1A*.

**Figure supplement 1—source data 2.** PDF containing *Figure 2—figure supplement 1A* and original scans of the relevant western blot analysis with highlighted bands and sample labels.

**Figure supplement 1—source data 3.** Original file for the colony formation assay in *Figure 2—figure supplement 1*.

**Figure supplement 1—source data 4.** PDF containing *Figure 2—figure supplement 1* and original scans of the relevant colony formation assay with highlighted bands and sample labels.

treatment with PARGi (*Figure 2B*). Consistently, moderate pADPr signaling was observed in PARG KO cells, but not in other cells (*Figure 2B*).

We reason that sensitivity to PARGi in PARG KO cells may depend on PARP1/2-dependent pADPr accumulation. To test this hypothesis, CellTiter-Glo and a colony formation assay were used to detect the sensitivity of WT, PARG KO, PARP1/2 DKO, and PARG/PARP1/2 TKO cells to PARGi. As expected, the sensitivity of PARG KO to PARGi was completely reversed in TKO cells, and DKO cells showed moderate resistance compared with WT cells (*Figure 2C*). Indeed, TKO cells completely rescued cell death induced by PARGi in PARG KO cells (*Figure 2—figure supplement 1C*). In addition, treatment with PARPi olaparib was able to reverse PARGi sensitivity in PARG KO cells (*Figure 2—figure supplement 1C*). These data together suggest that the cytotoxicity of PARGi in PARG KO cells was due to uncontrolled PARylation, which requires PARP1/2.

PARP1/2 catalyzes the pADPr reaction by transferring the ADR-ribose moiety of NAD $^+$ to the acceptor proteins with the release of NAM (*Fouquerel and Sobol, 2014*). Uncontrolled PARylation may lead to cytotoxicity, at least in part because of NAD $^+$ depletion (*Nagashima et al., 2020*; *Demin et al., 2021*; *Alano et al., 2010*). To determine whether cell death induced by PARGi in PARG KO cells is at least partially caused by NAD $^+$ depletion, we measured the relative NAD $^+$ level in WT and PARG KO cells treated with PARGi and used the nicotinamide phosphoribosyltransferase (NAMPT) inhibitor FK866 as a positive control. FK866 is a noncompetitive inhibition of NAMPT, a key enzyme involved in the regulation of NAD+ biosynthesis from its natural precursor NAM (*Hasmann and Schemainda, 2003*). As expected, the NAD $^+$ level decreased modestly over the treatment period in PARG KO cells, but this decrease was not as dramatic as those observed in cells treated with FK866 (*Figure 2—figure supplement 1D*). Moreover, treatment with FK866, which significantly decreased the NAD $^+$ level, did not lead to cell death (*Figure 2—figure supplement 1E*), suggesting that the decreased NAD+ level is not sufficient or the only reason for the cytotoxicity observed in PARG KO cells treated with PARGi. Furthermore, we treated cells with the NAD+ precursors nicotinamide mononucleotide (NMN) or NAM (*Nagashima et al., 2020*; *Liu et al., 2009*). As expected, NMN, NAM, and PARPi rescued the NAD $^+$ decrease in both non-treated and PARGi treated cells (*Figure 2D*). In addition, NMN and NAM were able to not only reduce pADPr accumulation induced by long-term PARGi treatment (*Figure 2E*), but also rescue cell lethality caused by PARGi treatment in both HEK293A-derived and HeLa-derived PARG KO cells (*Figure 2F*, *Figure 2—figure supplement 1F*). Together, these results suggest that uncontrolled PARylation in the absence of any dePARylation activities would lead to cytotoxicity, probably due to several mechanisms including apoptosis and are not limited to NAD+ depletion/reduction.

## Unexpected S phase pADPr signaling observed in PARG KO cells treated with PARGi

PARP1/2 are activated by DNA damage to generate pADPr signals. Under normal growth conditions, PARP1/2 activities may be low; therefore, it was extremely difficult to detect pADPr. Unexpectedly, pADPr signaling from normal proliferating cells without any exogenous DNA damage was detected in PARG KO cells after PARGi treatment (*Figure 2B*, *Figure 2—figure supplement 1B*), which was apparent in S phase cells, as detected by FACS analysis (*Figure 2B*). To further characterize pADPr and DNA damage signaling induced by PARGi or other DNA damaging agents, we used flow cytometry to detect both pADPr and γH2AX signaling (*Figure 3A*, *Figure 3—figure supplement 1A*). Interestingly, pADPr signaling was specifically detected in S phase in PARG KO cells treated with PARGi, while at the same time, DNA damage signaling only showed a slight increase or was not observed, as revealed by anti-γH2AX antibody (*Figure 3A*). On the other hand, MMS treatment led to increased pADPr signaling throughout the cell cycle (*Figure 3A*). However, it only led to an S-phase-specific increase in DNA damage signaling, that is γH2AX. Our interpretation of the MMS results is that MMS treatment would lead to DNA alkylation throughout the cell cycle, which is repaired by the BER pathway. BER creates single-strand nicks as repair intermediates, which are recognized by PARP1 and activate PARP1. Therefore, pADPr signaling can be detected in all cell cycle phases in PARG KO cells under this condition. However, single-strand nicks are not recognized by DNA damage checkpoint pathways. These nicks, when encountering replication forks, will be converted to DSBs and therefore activate DNA damage checkpoint kinases, resulting in increased γH2AX signaling specifically in S phase cells.

To further explore the relationship between pADPr and γH2AX signaling, we conducted a similar flow cytometry analysis in parental WT and PARG KO cells following treatments with different DNA damaging agents. Again, PARGi treatment in PARG KO cells led to an S-phase-specific increase in pADPr signaling; this unique pattern was not detected in response to any of the DNA damaging agents used in this study (*Figure 3—figure supplement 1A*). Our current working hypothesis is that PARG KO cells treated with PARGi may completely block any dePARylation reactions. The dramatic S-phase-specific pADPr signaling detected under this condition indicates that PARP1 is specifically activated in S phase cells.

During DNA replication, maturation of Okazaki fragments involves single-strand nicks on DNA, which is normally ligated by LIG1. These unligated Okazaki fragment intermediates could activate PARP1 to generate endogenous S phase pADPr (*Hanzlikova et al., 2018*.) It is possible that in such situations, the maturation of Okazaki fragments may require PARP1-dependent recruitment of XRCC1/LIG3 (*Hanzlikova et al., 2018*), which agrees with the results of early studies that indicate that both LIG1 and LIG3 are required for Okazaki fragment maturation (*Arakawa and Iliakis, 2015*). Of course, PARP1 may have other functions in the S phase, including DNA repair (*Hanzlikova and Caldecott, 2019*).

The S phase pADPr signaling observed in PARG KO cells after PARGi treatment was observed under conditions in which no or very limited DNA damage or replication stress–induced signaling was detected by anti-γH2AX antibody (*Figure 3A*) and S phase pADPr resulting from unligated Okazaki fragments were observed under similar conditions *Hanzlikova et al., 2018*; thus, we reasoned that unligated Okazaki fragments in normal proliferating cells may render this unexpected S phase pADPr signaling in PARG KO cells after PARGi treatment. Indeed, pre-treatment with emetine, which diminishes Okazaki fragments by its anti-protein synthetic activity (*Hanzlikova et al., 2018*; *Burhans et al., 1991*; *Lukac et al., 2022*), greatly inhibited S phase pADPr signaling in PARG KO cells (*Figure 3B*). Furthermore, mild inhibition of POLA1 with two inhibitors (i.e. adarotene and CD437) also abolished the PARGi induced S phase pADPr signaling in PARG KO cells (*Figure 3—figure supplement 1B*).

To further support S-phase-specific pADPr signaling resulting from unligated Okazaki fragments, we performed double thymidine block (DTB) and release experiments. Indeed, release into S phase is critical for the pADPr signaling observed in PARG KO cells treated with PARGi (*Figure 3C*). Moreover, the cytotoxicity of PARGi also requires S phase progression, since both control proliferating cells and double thymidine blocked and released cells died following PARGi treatment, while cells arrested by DTB failed to do so (*Figure 3D*). Additionally, mild inhibition of POLA1 partially rescued the cytotoxicity of PARGi in PARG KO cells (*Figure 3E*).

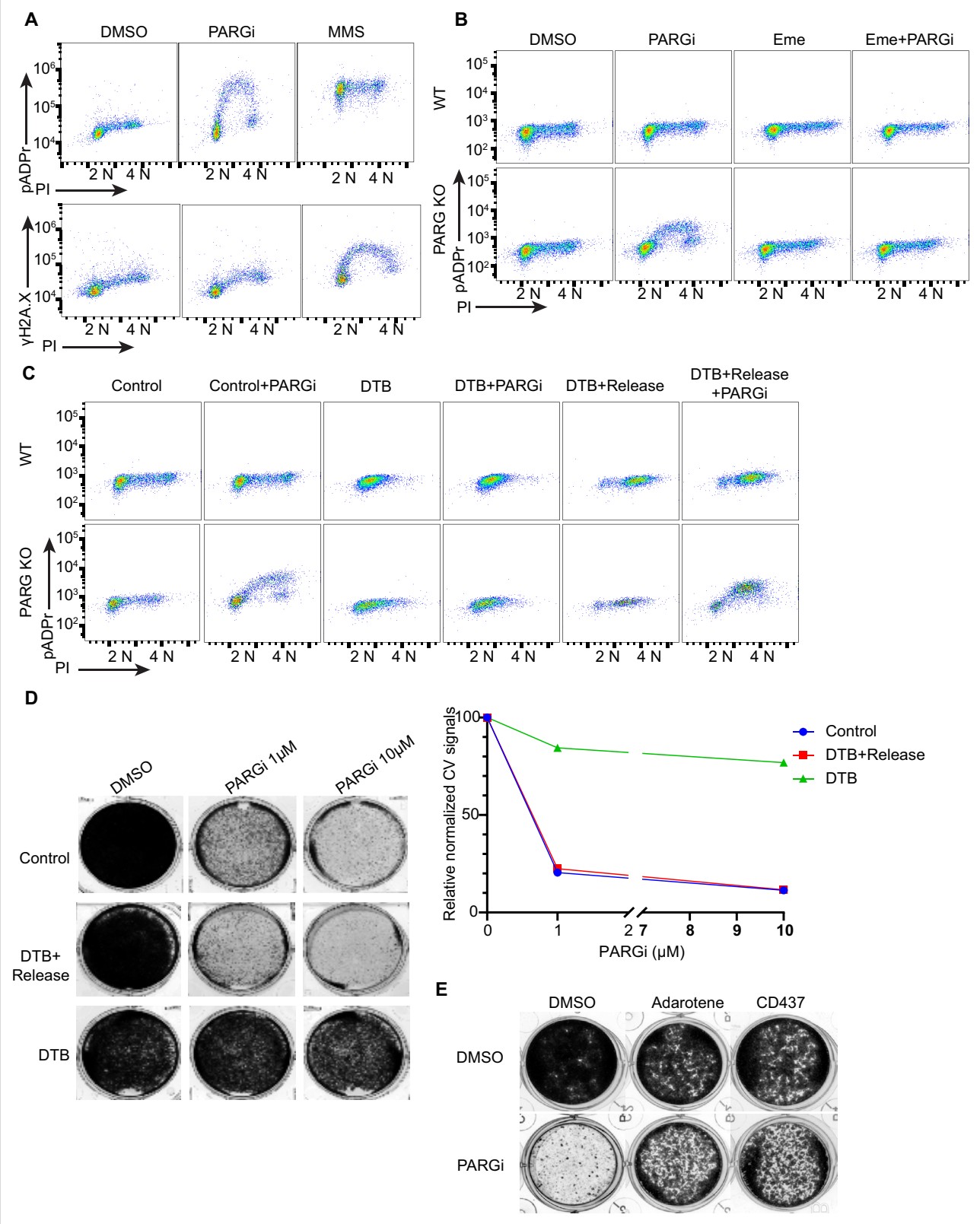

**Figure 3.** PARGi treatment induces S-phase-specific pADPr signaling in PARG KO cells. (**a**) HEK293A WT and PARG KO cells were treated with DMSO or 10 µM PARGi for 4 hr or 0.01% MMS for 30 min and then fixed and stained with anti-pADPr antibody or anti-γH2A.X antibody and PI. (**b**) HEK293A WT and HEK293A PARG KO cells were mock-treated or pre-treated with 2 µM emetine for 90 min and then treated with PARGi for an additional 4 hr. Cells were fixed and stained with anti-pADPr antibody and PI. (**c**) HEK293A PARG KO cells were synchronized with double thymidine block (DTB).

*Figure 3 continued on next page*

*Figure 3 continued*

Cells remained with DTB or were released from DTB, treated with 10 µM PARGi for 4 hr, and then fixed and stained with anti-pADPr antibody and PI. (**d**) Representative images and results (left) of clonogenic assays conducted using control cells and DTB synchronized or released HEK293A PARG KO cells treated with the indicated doses of PARGi for 7 days, and quantification of crystal violet staining assay (right). (**e**) Results of clonogenic assays were conducted in PARG KO cells with indicated treatment for 7 days (PARGi, 1 µM; adaratene, 200 nM; CD437, 800 nM).

The online version of this article includes the following source data and figure supplement(s) for figure 3:

**Source data 1.** Original file for the colony formation assay in *Figure 3*.

**Source data 2.** PDF containing *Figure 3* and original scans of the relevant colony formation assay with highlighted bands and sample labels.

**Figure supplement 1.** Treatment with DNA damaging agents did not induce S-phase-specific pADPr signaling.

## PARG KO cells show prolonged PARP1 chromatin binding

Following DNA damage, PARP1 detects and is activated by both SSBs and DSBs to initiate subsequent DNA repair (*Benjamin and Gill, 1980*; *Ikejima et al., 1990*). After that, PARP1 dissociates from DNA. If this does not occur, the persistent PARP1-DNA complexes could trap PARP1 on chromatin, which is one of the key characteristics of PARPi-mediated cytotoxicity (*Murai et al., 2012*). Given that the sensitivity of PARG KO cells to PARGi resulted from PARP1/2-dependent pADPr accumulation, which indicates an association and activation of PARP1/2 by DNA, we used a previously described trapping assay to measure the levels of chromatin-bound PARP1 (*Murai et al., 2012*; *Zhang et al., 2022*). Unexpectedly, the levels of chromatin-bound PARP1/2 increased in PARG KO cells, regardless of treatment, while the soluble fraction of PARP1 decreased compared with that in WT cells (*Figure 4—figure supplement 1A*).

To further explore the chromatin-bound PARP1, we performed similar soluble and chromatin fractionation and enrichment of PARylated proteins by Af1521 beads. Consistently, chromatin-bound PARP1 increased in PARG KO cells, regardless of whether these cells were treated with PARGi (*Figure 4A*). Moreover, PARylated PARP1 also increased in chromatin fraction. Interestingly, following PARGi treatment, PARG KO cells showed more pADPr levels than did WT cells in chromatin fraction, while similar pADPr levels were detected in soluble fractions (*Figure 4A*). Indeed, a recent study by Gogola and colleagues suggest that PARG depletion does not enhance PARP1 dissociation from DNA but prevents excessive PARP1 binding, on the basis of the results of a similar trapping assay and a laser-induced DNA damage assay for measuring PARP1 association (*Gogola et al., 2018*). Taken together, these results indicate that prolonged PARP1 chromatin binding or trapping occurs in PARG KO cells, which may contribute to the sensitivity of these cells to PARGi.

To further support the contribution of chromatin-bound PARP1 to PARGi sensitivity, we introduced either WT PARP1 or a trapping-deficient PARP1[del.p.119K120S] construction into TKO cells (*Figure 4—figure supplement 1B*). PARP1[del.p.119K120S] mutant is a trapping-deficient mutant, which leads to PARPi resistance (*Pettitt et al., 2018*; *Krastev et al., 2022*). Notably, expression of WT PARP1 led to dramatic PARGi sensitivity in TKO cells, while PARP1[del.p.119K120S] mutant only showed a slight increase in PARGi sensitivity (*Figure 4B*). Together, these data suggest that increased chromatin-bound and trapped PARP1 contributes, at least in part, to the sensitivity of these cells to PARGi.

To investigate the consequence of S phase pADPr and prolonged PARP1 chromatin binding in PARG KO cells, we again used flow cytometry to detect pADPr signaling in a time course following PARGi treatment. Consistent with the aforementioned data, S phase pADPr was detected following short PARGi treatment (i.e. 4 hr), while pADPr signaling was detected throughout the cell cycle following prolonged PARGi treatment, that is 24 or 48 hr (*Figure 4C*). Furthermore, SSBs were not detected following short PARGi treatment, as measured by an alkaline comet assay (*Figure 4—figure supplement 1C*), which may due to the sensitivity of this assay. However, prolonged PARGi treatment in PARG KO cells led to dramatic increase of SSBs detected by this assay (*Figure 4—figure supplement 1C*). Similarly, we observed S-phase-specific pADPr signaling with short PARGi treatment and pADPr signaling throughout the cell cycle with prolonged PARGi treatment in HeLa PARG KO cells, although significant S-phase pADPr was detected in control WT HeLa cells (*Figure 4—figure supplement 1D*), which was not observed in control WT 293 A cells (*Figure 4C*). The difference in pADPr signaling between HeLa and 293 A cells may due to a low level of PARG and/or high activity of PARP1 in HeLa cells, in which high–molecular weight smears were detected following PARGi and

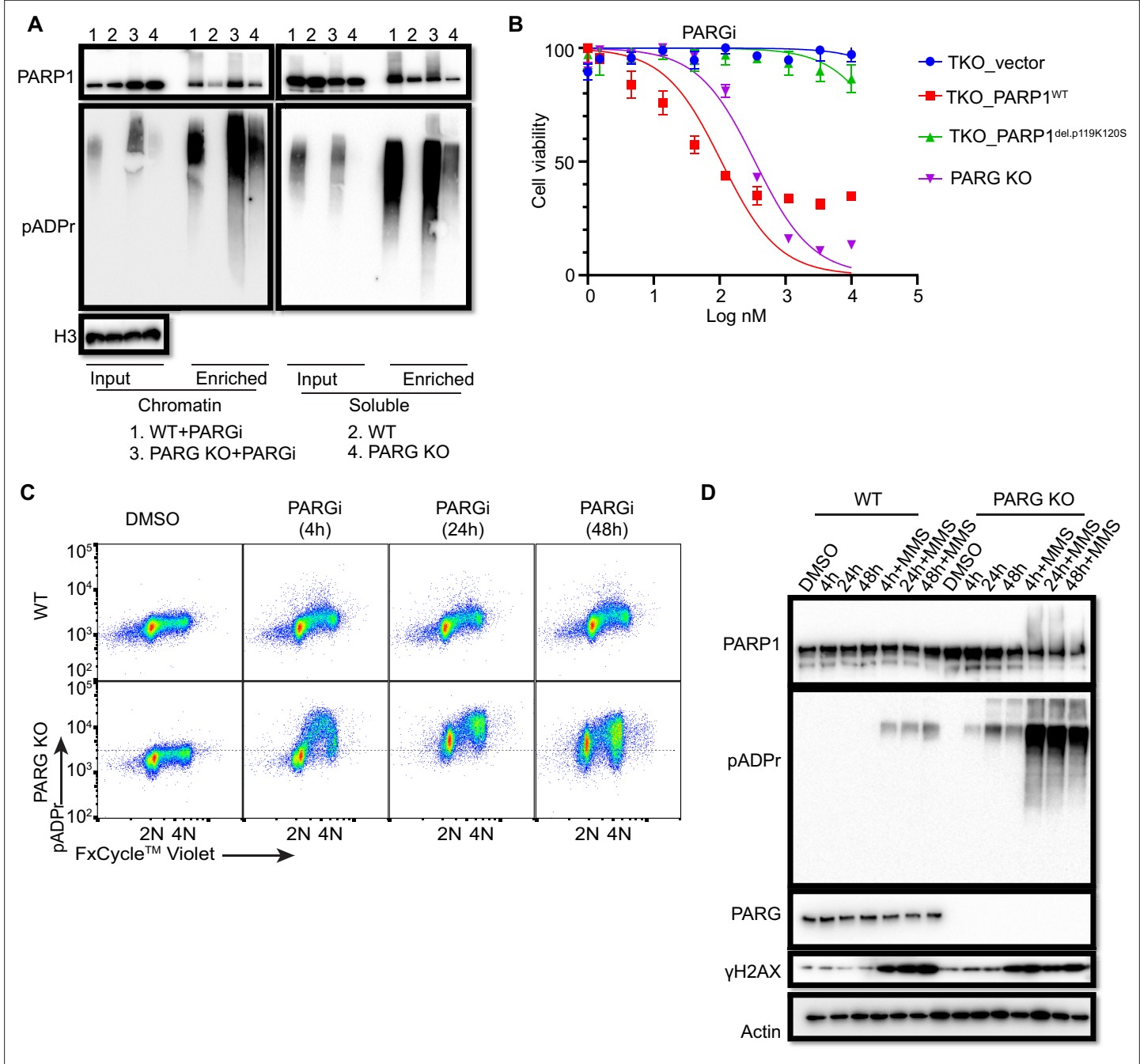

**Figure 4.** Prolonged PARGi treatment induces pADPr throughout the cell cycle and DDR in PARG KO cells. (**a**) Immunoblots of chromatin-bound PARP1 and PARylated proteins in HEK293A WT and PARG KO cells treated with PARGi (10 µM) for 4 hr. PARylated proteins were enriched by Af1521 beads. (**b**) Sensitivity of HEK 293 A PARG/PARP1/2 TKO and PARP1 reconstitution cells to PARGi. Cells were treated with different doses of PARGi for 72 hr, and cell viability was determined by the CellTiter-Glo assay. (**c**) Prolonged PARGi treatment induces pADPr throughout the cell cycle in PARG KO cells. HEK293A WT and PARG KO cells were treated with PARGi (10 µM) for the indicated time and then fixed and stained with anti-pADPr antibody and violet. (**d**) Immunoblotting of γH2A.X signals and other indicated proteins and modifications induced by prolonged PARGi +/-MMS treatment.

The online version of this article includes the following source data and figure supplement(s) for figure 4:

**Source data 1.** Original file for the western blot analysis in *Figure 4*.

**Source data 2.** PDF containing *Figure 4* and original scans of the relevant western blot analysis with highlighted bands and sample labels.

**Figure supplement 1.** Uncontrolled S phase pADPr accumulation eventually leads to DNA damage and cell death.

**Figure supplement 1—source data 1.** Original file for the western blot analysis in *Figure 4—figure supplement 1*.

*Figure 4 continued on next page*

*Figure 4 continued*

**Figure supplement 1—source data 2.** PDF containing *Figure 4—figure supplement 1* and original scans of the relevant western blot analysis with highlighted bands and sample labels.

**Figure supplement 1—source data 3.** Original images for alkaline comet assay in *Figure 4—figure supplement 1*.

MMS treatment (*Figure 4—figure supplement 1E*). Together, these data indicate that PARG regulates S-phase pADPr in normal proliferating cells.

To further confirm pADPr signaling observed by flow cytometry analysis, we detected pADPr and γH2AX by western blotting in HEK293A and HeLa cells and their corresponding PARG KO cells, while additional MMS treatment was included as a positive control (*Figure 4D*, *Figure 4—figure supplement 1F*). Expectedly, we only observed increased γH2AX signaling in PARG KO cells with prolonged PARGi treatment (48 hr of treatment in 293 A PARG KO cells and 24 and 48 hr of treatment in HeLa PARG KO cells) or those treated with additional MMS, indicating that pADPr accumulation leads to DNA damage throughout the cell cycle. The pADPr signaling in PARG KO cells pre-treated with PARGi was further enhanced in MMS-treated cells (*Figure 4D*, *Figure 4—figure supplement 1F*), likely due to increased SSBs; nevertheless, PARGi-mediated pADPr accumulation in PARG KO cells under condition without any exogenous DNA damage was sufficient to cause cytotoxicity. This situation is different from the hyperactivity and progressive inactivation of PARP1 in XRCC1-deficient cells after MMS treatment (*Demin et al., 2021*). Taken together, we speculate that inhibition of dePARylation in PARG KO cells treated with PARGi lead to pADPr accumulation and cell lethality.

## CRISPR screens reveal genes responsible for regulating pADPr signaling and/or cell lethality in WT and PARG KO cells

PARG KO and PARGi appear to have a synergic effect on cell viability, which depends on PARP1/2-dependent pADPr accumulation (*Figure 1*, *Figure 2*). Further, we showed that unligated Okazaki fragments were the likely source of SSBs, which leads to S phase pADPr signaling (*Figure 3*). Moreover, we found that uncontrolled S phase pADPr accumulation eventually leads to DNA damage, as revealed by anti-γH2AX antibody and alkaline comet assay (*Figure 4*). These data prompted us to further explore potential genes that may be responsible for controlling cell viability and pADPr signaling in control WT and PARG KO cells, with or without PARGi treatment. As we speculated above, there may be a potential second gene targeted by PARGi, which only becomes essential in the absence of PARG. We thus performed the outlined whole-genome CRISPR screens to address these questions (*Figure 5A*), similar to the studies published by us and others (*Wang et al., 2022*; *Condon et al., 2021*).

As for FACS-based CRISPR screening, we showed that PARP1 consistently appeared as the gene that, when depleted, led to reduced pADPr in all settings (*Figure 5B*, *Figure 5—figure supplement 1A*, *Supplementary file 1b-d*), which agrees with the dominant role of PARP1 in promoting PARylation in cells. The depleted genes that led to increased pADPr signaling were also similar in all of these settings (*Figure 5B*, *Figure 5—figure supplement 1A*); these include BER genes (i.e. POLB, XRCC1, LIG3, and CHD1L) and genes involved in DNA replication/Okazaki fragment maturation (i.e. RFCs, FEN1, and LIG1). Interestingly, ARH3 only appeared in PARG KO cells treated with PARGi (*Figure 5B*, *Figure 5—figure supplement 1A*), indicating that as anticipated ARH3 may serve as a backup dePARylating enzyme.

We further investigated whether the aforementioned genes that regulated pADPr signaling would show strong synthetic lethality with PARGi in both PARG KO cells (*Figure 5C*, *Supplementary file 1e*) and WT cells (*Figure 5D*, *Supplementary file 1f*). To identify the most confident hits, we used a stringent false discovery rate (FDR) threshold of 0.05 (*Zimmermann et al., 2018*). Consistently, PARP1 was listed as the most resistant gene with PARGi in PARG KO cells (*Figure 5C*). Interestingly, ARH3 appeared as the top gene in PARG KO cells treated with PARGi (*Figure 5C*), while other dePARylation enzymes (e.g. MARCROD1/2 andTARG1/OARD1) did not show any synthetic lethal effect in these screens. These data agree with the results of FACS-based screens and suggest that ARH3 is a backup enzyme in the absence of PARG following PARGi treatment. Indeed, loss of ARH3 further sensitized PARG KO cells to PARGi (*Figure 5—figure supplement 1B*). In addition, HPF1, another gene/protein involved in PARylation regulation and licensing serine mono-ADP-ribosylation by PARP1/2

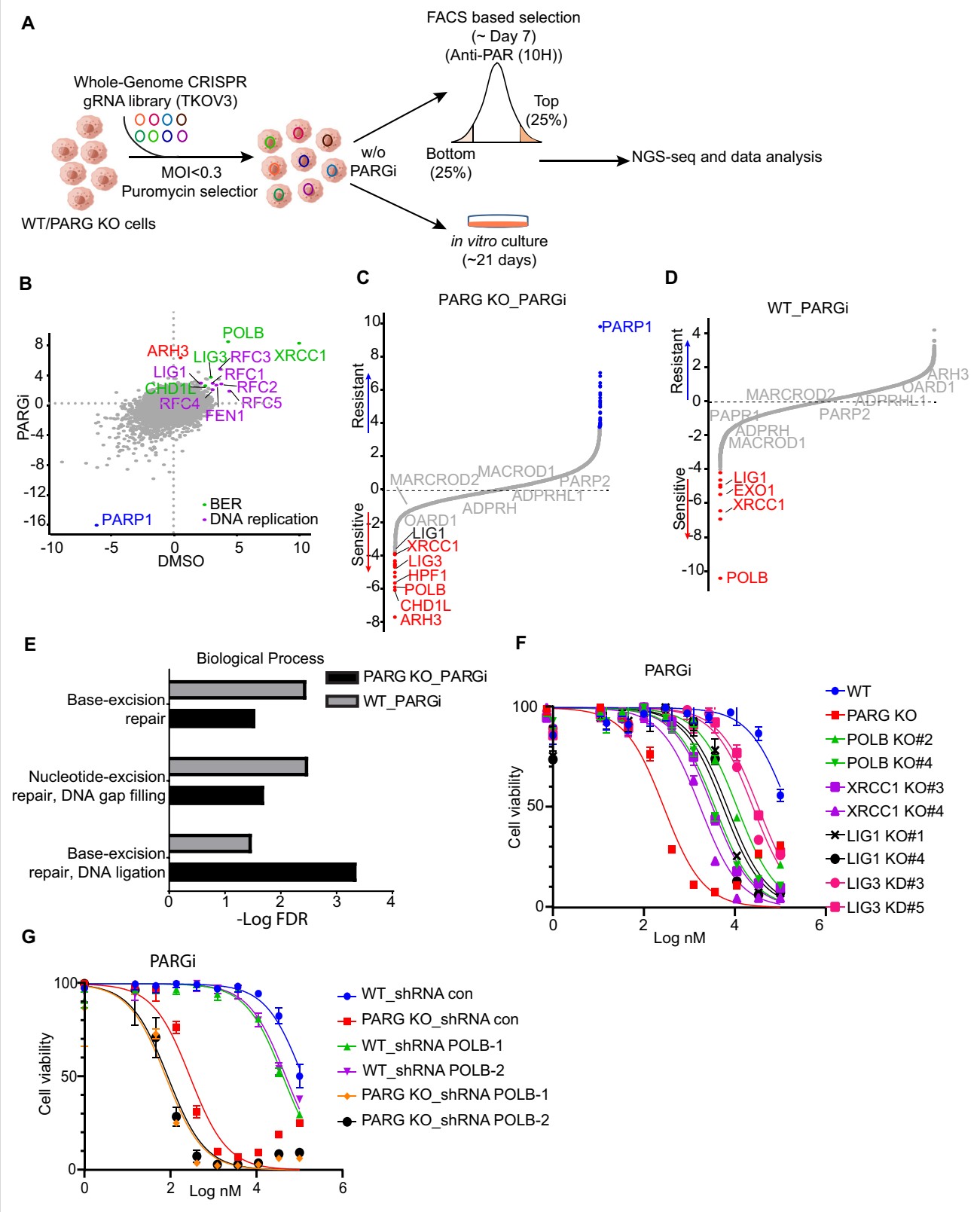

**Figure 5.** CRISPR screening identifies regulators of pADPr and cell viability. (**a**) Workflow of whole-genome CRISPR screens. For FACS-based CRISPR screening, 5 days after puromycin selection, cells were treated with PARGi (10 μM) for 4 hr and then stained with anti-pADPr antibody and sorted with flow cytometry. Cells with strong signals (top 25%, TOP) and weak signals (bottom 25%, BOT) were selected. The sgRNAs from these cells were then sequenced and analyzed. For cell viability screening, cells were treated with or without PARGi for 21 days before collection. (**b**) Scatter plot of DrugZ

*Figure 5 continued on next page*

Figure 5 continued

scores of PARG KO cells treated with or without PARGi treatment. The genes in the same pathway were marked with specific colors. A positive score indicates an enhanced pADPr signal, while a minus score indicates a decreased pADPr signal. (**c** and **d**) Ranking of PARGi co-essential genes on the basis of a DrugZ analysis of the results of CRISPR/Cas9 screens performed with Toronto Knock Out Library (version 3) in HEK293A PARG KO cells and HEK239A cells. (**e**) Analysis of biological processes of PARGi co-essential genes identified in HEK293A PARG KO cells and HEK239A cells. (**f**) HEK293A WT cells, PARG KO, POLB KO, LIG1 KO, XRCC1 KO, and LIG3 knockdown cells were treated with different doses of PARGi for 72 hr. Cell viability was determined by the CellTiter-Glo assay. (**g**) The cell Viability of HEK293A WT and PARG cells under POLB knockdown to PARGi. Cells were treated with different doses of PARGi for 72 hr.

The online version of this article includes the following source data and figure supplement(s) for figure 5:

**Source data 1.** NormZ score of FACS-based TKOv3 library screen conducted with PARG KO cells with or without PARGi.

**Source data 2.** NormZ score of cell viability–based TKOv3 library screen conducted with HEK293A cells or PARG KO cells treated with PARGi.

**Figure supplement 1.** Proteins involved in pADPr regulation contribute to PARGi sensitivity.

**Figure supplement 1—source data 1.** NormZ score of FACS-based TKOv3 library screen conducted with HEK293A cells treated with PARGi.

**Figure supplement 1—source data 2.** Original file for the western blot analysis in *Figure 5—figure supplement 1*.

**Figure supplement 1—source data 3.** PDF containing *Figure 5—figure supplement 1* and original scans of the relevant western blot analysis with highlighted bands and sample labels.

---

(*Gibbs-Seymour et al., 2016*), was among the top synthetic lethality genes in PARG KO cells treated with PARGi, but not in WT cells (*Figure 5C, D*, *Figure 5—figure supplement 1C*).

Thus, our CRISPR screens revealed that genes involved in PARylation regulation (ARH3 and HPF1), BER (i.e. POLB, XRCC1, LIG3, and CHD1L), and DNA replication/Okazaki fragment maturation (i.e. RFCs, FEN1, and LIG1) are responsible for pADPr signaling and/or cell lethality in WT and PARG KO cells.

To further explore the involvement of these aforementioned genes, we compared synthetic lethality genes in PARG KO cells with those in WT cells treated with PARGi. Interestingly, two BER key genes, POLB and XRCC1, showed strong synthetic lethality with PARGi in both PARG KO cells (*Figure 5C*) and WT cells (*Figure 5D*), which is consistent with the results of a previous report that POLB-deficient cells are sensitive to PARGi (*Ali et al., 2021*). Moreover, the GO analysis revealed that the BER pathway displays synthetic lethality in both PARG KO cells and WT cells following PARGi treatment (*Figure 5E*). Importantly, LIG1 and LIG3, the two ligases that are directly involved in Okazaki fragment ligation, were also listed as synthetic lethality genes (*Figure 5C, D*). To further validate these data, we created several KO/KD cell lines, including KO of BER genes (i.e. POLB and XRCC1) and genes involved in Okazaki fragment ligation (i.e. LIG1 and LIG3; *Figure 5—figure supplement 1D*). Consistent with previous reports, XRCC1 loss destabilizes LIG3 (*Figure 5—figure supplement 1D*; *Lee et al., 2009*; *Caldecott et al., 1994*). Notably, loss of genes involved in Okazaki fragment ligation induced S phase pADPr signaling, just like PARG KO cells following PARGi treatment, while loss of BER genes resulted in pADPr signaling throughout the cell cycle (*Figure 5—figure supplement 1E*). These results agree with the aforementioned results that unligated Okazaki fragments are the likely source of SSBs, which induces S-phase-specific pADPr signaling. Also consistent with CRISPR screening data, loss of genes involved in Okazaki fragment ligation or BER resulted in increased sensitivity to PARGi (*Figure 5E*). Furthermore, we knocked down POLB in WT and PARG KO cells (*Figure 5—figure supplement 1F*), which resulted in increased sensitivity to PARGi in both WT and PARG KO cells (*Figure 5G*). Together, our data indicate that unligated Okazaki fragments induce S-phase-specific pADPr, while BER intermediates lead to pADPr throughout the cell cycle; deficiency in either of these events would increase cytotoxicity following PARGi treatment.

## PARG expression is a potential biomarker for PARGi-induced cytotoxicity

We showed that cells with no detectable full-length PARG expression are extremely sensitive to PARG inhibition (*Figure 1*, *Figure 1—figure supplement 1*). Thus, we are interested in determining whether PARG expression would correlate with PARGi sensitivity. An early study examined a panel of ovarian cancer cell lines and reported PARGi sensitivity in a subset of these cell lines (*Pillay et al., 2019*). We compared PARG expression in PARGi sensitive and resistant cell lines and found that PARG expression was significantly reduced in PARGi sensitive cell lines (*Figure 6A*). To support this result, we knocked

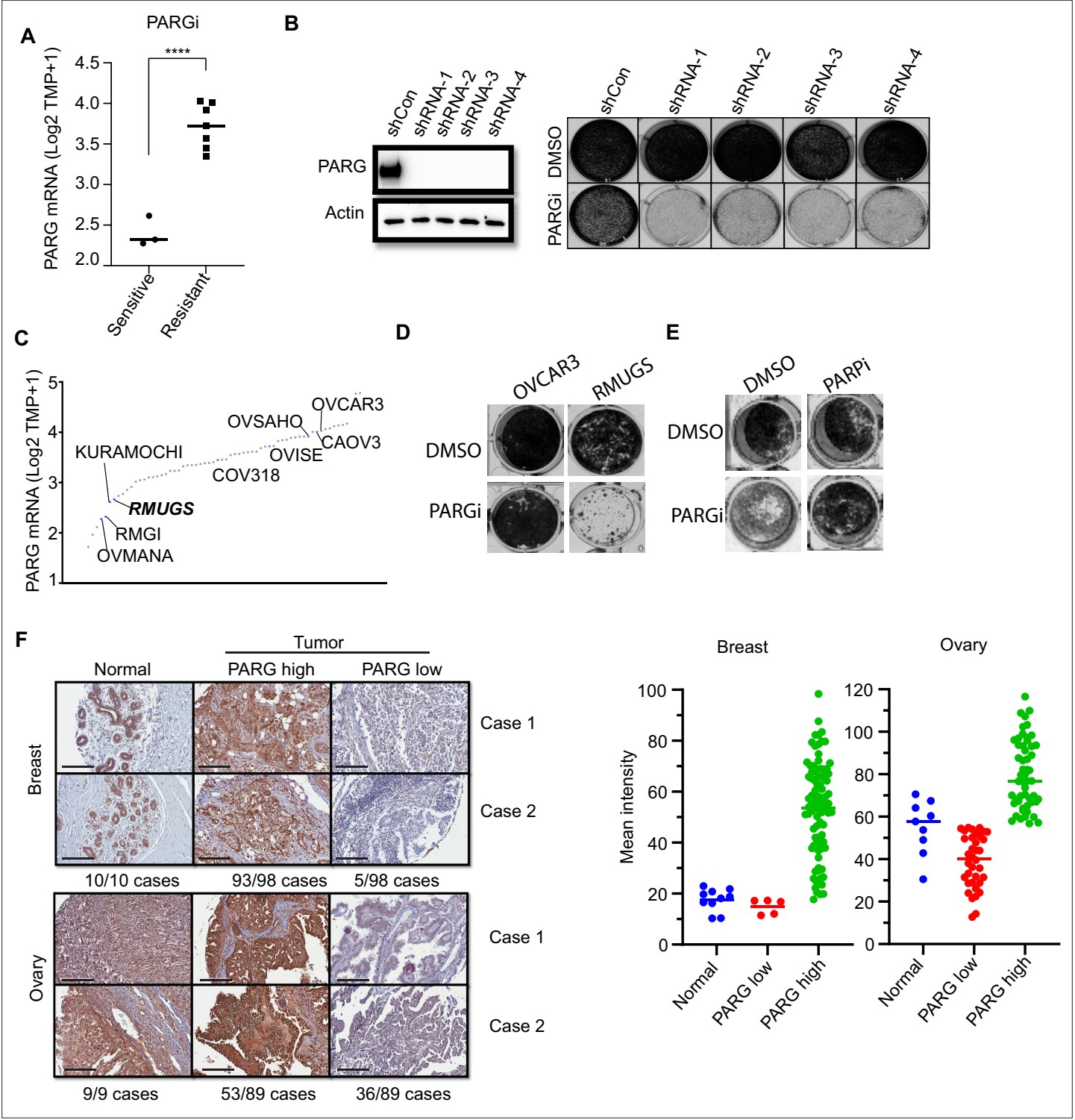

**Figure 6.** PARG expression is a potential marker for PARGi sensitivity. (**a**) PARG mRNA level comparison between sensitive (RMGI, KURAMOCHI, and OVMANA) and resistant cells (COV362, COV318, OV56, OVISE, OVSAHO, CAOV3, and OVCAR3) from Pillay's work on the basis of CCLE data. (**b**) Clonogenic assay results of control and PARG knockdown HeLa cells with PARGi (2 μM) treatment for 7 days. shRNA knockdown efficiency was confirmed by an immunoblot of PARG. (**c**) Ranked PARG expression level in ovarian cancer cell lines based on the CCLE database. The sensitive and resistant cells from Pillay's work were labeled. (**d**) Clonogenic assay results of OVCAR3 and RMUGS treated with or without PARGi (2 μM). (**e**) Clonogenic assay results of RMUGS treated with PARGi (2 μM), PARPi (2 μM), or both. (**f**) Left: Representative images of PARG IHC staining in breast and ovarian tissues and tumor samples to determine the PARG expression level. The summary is listed at the bottom. Scale bar, 200 μm. Right: The scatter plot of mean immunostaining intensity of PARG in each sample. The mean of each group was plotted.

*Figure 6 continued on next page*

*Figure 6 continued*

The online version of this article includes the following source data and figure supplement(s) for figure 6:

**Source data 1.** Original file for the western blot analysis and colony formation assay in *Figure 6*.

**Source data 2.** PDF containing *Figure 6* and original scans of the relevant western blot analysis and colony formation assay with highlighted bands and sample labels.

**Figure supplement 1.** HR deficiency renders cells sensitive to PARGi.

**Figure supplement 1—source data 1.** Original file for the western blot analysis and colony formation assay in *Figure 6—figure supplement 1*.

**Figure supplement 1—source data 2.** PDF containing *Figure 6—figure supplement 1* and original scans of the relevant western blot analysis and colony formation assay with highlighted bands and sample labels.

**Figure supplement 2.** PARG loss is a robust marker of PARGi sensitivity.

**Figure supplement 2—source data 1.** Original file for the western blot analysis and colony formation assay in *Figure 6—figure supplement 2*.

**Figure supplement 2—source data 2.** PDF containing *Figure 6—figure supplement 2* and original scans of the relevant western blot analysis and colony formation assay with highlighted bands and sample labels.

down PARG by shRNA in HeLa cells and showed that loss or further reduction of PARG led to dramatic sensitivity to PARGi (*Figure 6B*). To further test our hypothesis, we ranked the PARG mRNA level in several ovarian cancer cell lines using the CCLE database (*Figure 6C*). We identified an ovarian cancer cell line, RMUGS, which has low PARG expression, similar to that observed in another known PARGi-sensitive cell line, KURAMOCHI. Notably, RMUGS cells were considered as potential PARGi-resistant cells based on an earlier report (*Pillay et al., 2019*). However, we showed that RMUGS cells were very sensitive to PARGi treatment (*Figure 6D*), and the sensitivity of these cells to PARGi could be reversed by PARPi treatment (*Figure 6E*). These data indicate that PARG expression may dictate the sensitivity of tumor cells to PARGi-based therapy.

An early study reported that loss of PARG expression resulted in PARPi resistance (*Gogola et al., 2018*). They demonstrated that PARG is frequently lost in acquired PARPi-resistant mouse mammary tumors and further revealed that PARG depletion occurs in triple-negative breast and ovarian cancer. Another study showed that 60 of 274 (22%) of human ovarian tumors have low PARG expression (*Ali et al., 2021*). Thus, PARGi can be potentially used to treat these PARPi-resistant cancers, especially those with low PARG expression. We further performed an IHC assay to detect PARG expression in breast and ovarian cancer TMA samples. The IHC assay was established with the use of our xenograft tumors derived from HeLa and HeLa PARG KO cells (data not shown). As shown in *Figure 6F*, PARG expression was detected in normal ovarian and breast tissues. PARG expression was also detected in a majority or most of ovarian and breast tumor samples (*Figure 6F*). Interestingly, about 5% to 40% of these breast or ovarian cancer tumor samples showed low expression levels of PARG, some even showed no detectable level of PARG, compared with those in normal tissue samples (*Figure 6F*). These data suggest that PARG downregulation occurs in breast and ovarian cancers. PARGi may be a promising strategy for the treatment of these cancers with low PARG expression.

In early studies, HR-deficient cells showed increased sensitivity to PARG inhibition (*Fathers et al., 2012*; *Gravells et al., 2017*; *Chen and Yu, 2019*; *Jain et al., 2019*). Similarly, we observed PARGi-induced cytotoxicity in RPE1 Flag-Cas9 TP53/BRCA1 DKO cells, but not in control cells (*Figure 6—figure supplement 1A*). Moreover, combining PARPi and PARGi did not reveal any additive effect (*Figure 6—figure supplement 1A*). Furthermore, we created inducible BRCA1 depletion cells using the auxin-inducible degron (mAID) tag. As shown in *Figure 6—figure supplement 1B*, HR-proficient cells showed normal sensitivity to PARGi, while HR deficiency by the loss of BRCA1 conferred modest sensitivity to PARGi. The PARGi sensitivity in BRCA1 depletion cells was reversed to normal by the restoration of HR due to 53BP1 loss. Consistent with the results obtained in RPE1 cells, the PARPi + PARGi combination did not lead to any further change in cytotoxicity. While these results indicate that PARGi can be used to target HR-deficient cancers, the effect is quite modest. Moreover, PARGi treatment did not show any effect on PARPi-resistant cells due to restoration of HR mediated by 53BP1 loss.

To further investigate the potential contribution of HR deficiency and/or PARG loss to PARGi sensitivity, we knocked down PARG by shRNA in these inducible BRCA1 depletion cells with or without 53BP1 KO (*Figure 6—figure supplement 2A*). As expected, PARG loss is the main driver of PARGi sensitivity; regardless whether these cells are HR-proficient, HR-deficient, or HR-restored because

of 53BP1 loss (*Figure 6—figure supplement 2B*). Our data therefore strongly suggest that PARG expression is the major determinant of PARGi sensitivity.

## PARG is essential for cell survival

Our results suggested that cells with no detectable full-length PARG expression or low expression of PARG are extremely sensitive to PARG inhibition (*Figure 1*, *Figure 6*). As aforementioned, the question is why PARG KO cells would be sensitive to PARGi. To address this question, we first performed whole-genome CRISPR screening. However, the potential second target was not revealed by our CRISPR screening data (*Figure 5*). The second possibility is that our PARG KO cells are not complete PARG KO cells. These cells may have residual PARG expression or activity and only cells with very low PARG expression are sensitive to PARGi, as shown in *Figure 6*. Indeed, two independent Parg knockout mice have been created by targeting early exons of PARG (*Koh et al., 2004*; *Cortes et al., 2004*). While one group of Parg knockout mice showed embryonic lethality and cells derived from these mice only survived in the presence of PARPi (*Koh et al., 2004*), the other Parg knockout mice did not display any major phenotypes, probably because truncated Parg and Parg activity were still detected (*Cortes et al., 2004*). We favor the second possibility, especially since the PARG catalytic domain is at its C-terminus.

To test this possibility, we first used two independent antibodies that recognize the C-terminus of PARG to detect potential residual PARG isoforms or spliced variants (*Figure 7—figure supplement 2A*). Unfortunately, besides full-length PARG, these antibodies also recognized several other bands, some of them were reduced or absent in PARG KO cells, others were not. Thus, we could not draw a clear conclusion which functional isoform/truncated form was expressed in our PARG KO cells. Then, we used two additional gRNAs which target respectively the beginning of the catalytic domain and the sequence around the catalytic site (*Figure 7A*). Interestingly, we were only able to obtain viable clones with no detectable PARG in the presence of PARPi in either 293 A or HeLa cells, which we named as PARG complete/conditional KO cells (cKO) (*Figure 7B*). Indeed, these cKO cells could not survive without PARPi (*Figure 7C*). To further investigate whether cell lethality is associated with PARG activity in these cells, we reconstituted PARG cKO cells with WT PARG or a PARG catalytic-inactive mutant. Consistent with previous result (*Figure 1*, *Figure 1—figure supplement 1*), WT PARG not only reduced pADPr level but also rescued cell lethality due to PARG loss, while the catalytic-inactive mutant PARG failed to do so (*Figure 7D*).

The data above indicate that our PARG KO cells have residual PARG activity and treatment with PARGi in these cells further decreases PARG activity, which mimic the complete loss of PARG as that in PARG cKO cells. To further test this hypothesis, we measured the relative PARG dePARylation activity in whole cell lysates (WCL) prepared from different cell lines (*Figure 7—figure supplement 2C*). Consistent with our working hypothesis, residual PARG activity was detected in WCL prepared from PARG KO cells, while PARG activity was barely detected or absent in WCL isolated from PARG cKO cells. Please note that we incubated cell lysates with substrates overnight to evaluate the maximum level of pADPr hydrolysis, that is PARG activity, we were able to detect in these assays. It is very likely that the PARG activity in PARG KO cells was much lower than that indicated in *Figure 7—figure supplement 2C*, due to saturation of signals for lysates isolated from wild-type cells. Thus, the data presented here may underestimate the reduction of PARG activity in PARG KO cells. Nevertheless, these data indicate the residual PARG activity in PARG KO cells, which is absent in PARG cKO cells. Furthermore, the PARG activity was further inhibited by PARGi in a dose-dependent manner in WCLs prepared from both WT and PARG KO cells (*Figure 7—figure supplement 2D*, *Figure 7—figure supplement 2E*). Notably, the PARG activity was consistently lower in WCL prepared from PARG KO cells when compared with that in control wild-type cells in the presence of a wide range of PARGi concentrations. However, a significant fraction of PARG activity still existed in WCL prepared from wild-type cells even at the highest concentration of PARGi used in this study (*Figure 7—figure supplement 2E*). These data indicate residual PARG activity in PARG KO cells likely accounts for the survival of these cells. Moreover, further inhibition of this activity by PARGi likely leads to cell lethality.

We showed above that $NAD^+$ precursors (NAM and NMN) were able to rescue cytotoxicity of PARGi in PARG KO cells, probably by $NAD^+$ depletion and also that these precursors can also inhibit PARP1/2 activities. Consistently, PARG cKO cells could survive in the presence of $NAD^+$ precursors (NAM and NMN; *Figure 7E*). Taken together, PARG is an essential gene and our PARG KO cells

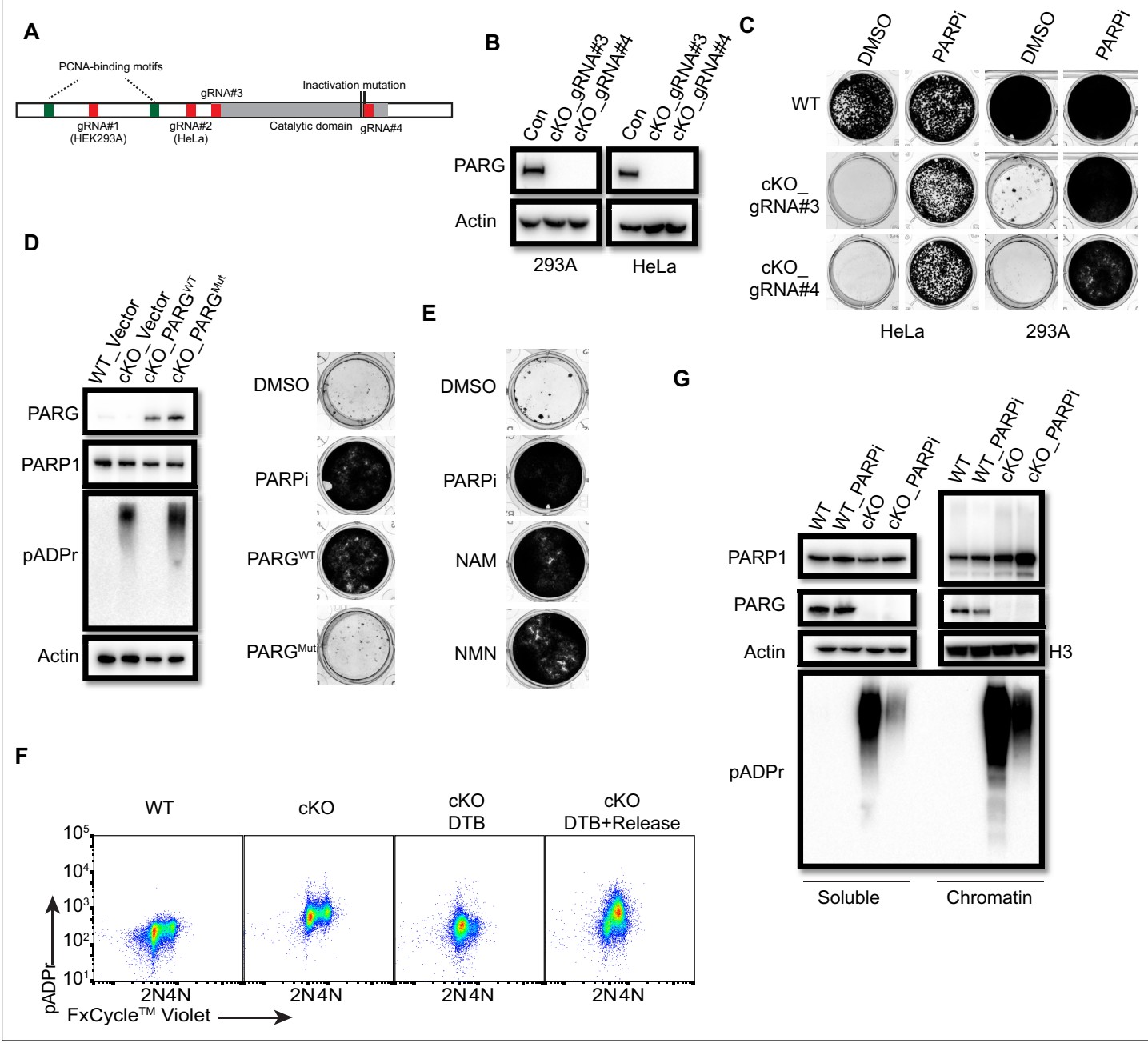

**Figure 7.** PARG is essential for cell survival. (**a**) Diagram of full-length PARG was presented with the indicated gRNAs (gRNA#3 and 4) which target different regions in the C-terminal catalytic domain. The boundary of the catalytic domain was depicted based on Uniprot annotation. gRNA#1 and gRNA#2 were used previously to generate the aforementioned HEK293A- and HeLa-derived PARG KO cells, respectively; while gRNA#3 and gRNA#4 were used to generate PARG complete/conditional knockout (cKO) in the presence of olaparib in HEK293A and HeLa cells. (**b**) Immunoblotting was conducted to confirm the loss of PARG in PARG cKO cells derived from HEK293A and HeLa cells, which were cultured in the presence of 100 nM olaparib. (**c**) Clonogenic assay results of WT and PARG cKO cells treated with or without PARPi (100 nM) for 7 days. (**d**) Left: The immunoblots to confirm reconstitution with WT PARG or catalytic inactivation PARG in HEK293A PARG cKO cells. Right: Results of clonogenic survival assay with HEK293A PARG cKO cells reconstituted with WT or catalytic inactivation mutant of PARG for 7 days. (**e**) Representative clonogenic results conducted in HEK293A PARG cKO cells treated with NAM (100 µM) or NMN (1 mM) for 7 days. (**f**) HEK293A PARG cKO cells were synchronized with double thymidine block (DTB). Cells remained with DTB or were released from DTB for 4 hr, and then fixed and stained with anti-pADPr antibody and FxCycle Violet dye. (**g**) Immunoblots of soluble and chromatin-bound PARP1 and pADPr levels in HEK293A WT and PARG cKO cells treated with DMSO or olaparib (10 µM) for 2 hr.

The online version of this article includes the following source data and figure supplement(s) for figure 7:

*Figure 7 continued on next page*

*Figure 7 continued*

**Source data 1.** Original file for the western blot analysis and colony formation assay in *Figure 7*.

**Source data 2.** PDF containing *Figure 7* and original scans of the relevant western blot analysis and colony formation assay with highlighted bands and sample labels.

**Figure supplement 1.** KO cells were validated by DNA sequencing.

**Figure supplement 2.** DePARylation activity of PARG is essential for cell survival.

**Figure supplement 2—source data 1.** Original file for the western blot in *Figure 7—figure supplement 2A*.

**Figure supplement 2—source data 2.** PDF containing *Figure 7—figure supplement 2A* and original scans of the relevant western blot analysis with highlighted bands and sample labels.

treated with PARGi can mimic the complete loss of PARG activity. These data agree with the genetically engineered mouse models as we discussed above (*Koh et al., 2004*).

We showed that S phase pADPr signals were specifically detected in PARG KO cells treated with PARGi, which likely originated from unligated Okazaki fragments (*Figure 3*). As expectedly, we observed S-phase-specific pADPr in PARG cKO cells under double-thymidine block and release condition (*Figure 7F*). Moreover, the levels of chromatin-bound PARP1 and pADPr signal, which reflects PARylated PARP1 and/or other PARP1 substrates, increased in PARG cKO cells (*Figure 7G*). Treatment with PARPi decreased pADPr signals but increased chromatin-bound PARP1 (*Figure 7G*), which is consistent with the results obtained in PARG KO cells (*Figure 4A*). Together, PARG, as an essential gene, suppresses initial S phase pADPr signaling and PARP1 chromatin accumulation, which eventually lead to cell death caused by uncontrolled pADPr accumulation throughout cell cycle.

## Discussion

Aberrant accumulation of pADPr and/or NAD$^+$ exhaustion disrupts multiple cell processes, including DNA repair, replication stress response, and transcription regulation; therefore, it is highly toxic and eventually leads to cell death (*Prokhorova et al., 2021*; *Mashimo et al., 2013*; *Demin et al., 2021*; *Alano et al., 2010*; *Adamowicz et al., 2021*). Although ADP-ribose hydrolases, especially PARG, have a protective effect against excessive PARP1 engagement, it remains unclear when and how PARG activity is engaged during normal cell proliferation. In this study, we showed that the major function of PARG is to regulate S-phase-specific pADPr by PARP1, which is activated by unligated Okazaki fragments. When PARG activity is inhibited, these uncontrolled pADPr would eventually result in DNA damage and cell death. Using unbiased FACS- and cell viability-based genome-wide CRISPR screens, we uncovered additional pADPr modulators, which include the genes involved in BER (POLB, XRCC1, and LIG3), Okazaki maturation (LIG1 and FEN1), ARH3, and HPF1. More importantly, we determined that PARG expression is a critical biomarker for PARGi sensitivity. Furthermore, we showed that PARG is an essential gene by suppressing pADPr accumulation. Taken together, our study uncovers a normal function of PARG in proliferating cells: it removes pADPr generated by PARP1 at unligated Okazaki fragments during DNA replication. Of course, this normal function of PARG is also required under conditions of excessive PARP1 engagement because of defects in Okazaki fragment maturation and/or BER.

An important lesson we learned from this study is that it is challenging to confirm the complete knockout of a gene of interest. As we reported here, we initially used two independent sgRNAs, which in theory target all active forms of PARG, and created KOs in two cell lines. Additionally, we validated our KO clones by western blotting, DNA sequencing and MMS-induced PARylation. Despite these efforts and our inability to detect full-length PARG in our KO clones, these PARG KO cells still express one or more active fragments of PARG, probably due to alternative splicing and/or alternative ATG usage. In this study, we were able to measure PARG activity directly in cell lysates using a biochemical assay, which supports our working hypothesis. However, such assay is not available for many enzymes or proteins without known enzymatic activities. Thus, one should be cautious when drawing conclusions based on KOs, especially conclusions such as genetic interactions that rely on multiple KOs.

PARylation is a reversible post-translational modification. In this study, we showed that timely and efficient dePARylation is critical for cell survival. Thus, both PARylation and dePARylation play important functions in the cell. Particularly, dePARylation appears to be indispensable, since it is required for cell

survival. The successful clinical application of PARylation inhibition, such as PARPis for the treatment of BRCA-deficient cancers, raises the possibility of targeting dePARylation as an alternative strategy, especially for the treatment of PARPi-resistant tumors. Given that PARG accounts for more than 90% of dePARylation activity (*Davidovic et al., 2001*; *Min and Wang, 2009*), selective PARG inhibitors are being developed as potential anti-cancer agents. Although defective HR and DDR have been shown to be synthetic lethal with PARG inhibition, comprehensive investigations of cellular response to PARG inhibition have not yet been conducted. In this study, we showed that unexpectedly, PARG is the top synthetic lethality gene in the CRISPR screening with our homemade DDR sgRNA library, which we further validated in HEK293A and HeLa PARG KO cells (*Figure 1B*, *Figure 1—figure supplement 1D*). Our PARG KO cells plus PARGi mimic complete loss of PARG activity (*Figure 7—figure supplement 2*), which provides is an unique opportunity to study the major function of PARG in proliferating cells. These data were further confirmed with the use of PARG cKO cells we generated (*Figure 7*).

PARP1 plays important roles during normal S phase, especially at DNA replication forks and two-ended DSBs (*Hanzlikova and Caldecott, 2019*). More recently, unligated Okazaki fragments were identified as the source of SSBs that activate PARP1 in normal proliferating cells (*Hanzlikova et al., 2018*). Moreover, unligated Okazaiki fragments, on which PARP1 may be trapped, can be converted into structures that result in synthetic lethality in HR-defective cancer cells (*Cong et al., 2021*). PARG is also enriched at DNA replication forks through pADPr- and PCNA-dependent mechanisms (*Mortuse-wicz et al., 2011*; *Kaufmann et al., 2017*; *Dungrawala et al., 2015*). In addition, PARG may protect DNA replication forks during normal and damage-treated S-phase (*Shirai et al., 2013b*; *Houl et al., 2019*; *Ray Chaudhuri et al., 2015*). Interestingly, the identification of unligated Okazaki fragments as the source of endogenous pADPr in normal proliferating cells was only noticeable under short-term PARG inhibition. Treatment with another recently developed PARG inhibitor COH34, also led to increased PARylation at replication sites in S phase cells (*Chen and Yu, 2019*). These previous reports agree with the data presented in this study. Indeed, we showed robust and specific PARP1/2-dependent S phase pADPr in PARG KO cells treated with PARGi (*Figure 3A*, *Figure 3—figure supplement 1*). We further confirmed that the S-phase-specific pADPr requires normal S phase progression and the generation of Okazaki fragments (*Figure 3B–C*, *Figure 3—figure supplement 1B*). Moreover, the cytotoxicity of PARGi requires S phase progression (*Figure 3D–E*). In addition, PARG KO cells displayed more chromatin-bound PARP1 and PARylated PARP1 (*Figure 4A*, *Figure 4—figure supplement 1A*). Enhanced pADPr levels were observed in the chromatin fraction of PARG KO cells under PARGi treatment (*Figure 4A*). One possible explanation is that in PARG KO cells treated with PARGi, chromatin-bound or trapped PARP1 may not be removed appropriately or efficiently, which further prevents Okazaki fragment ligation/maturation that eventually results in cell lethality. However, the mechanisms underlying chromatin accumulation of PARP1 and PARylated PARP1 in PARG KO cells remain to be further elucidated.

Additionally, we do not yet know all the substrates of PARG in S phase cells. We speculate that PARP1 is one of the major PARG substrates in S phase cells. As mentioned above, chromatin-bound PARP1 as well as PARylated PARP1 increased in PARG KO cells. Moreover, PARP1 depletion was able to rescue cell lethality in PARG cKO cells or PARG KO cells treated with PARGi. Of course, PARG may have additional substrates besides PARP1 which are required for its roles in S phase progression. Precisely how PARG regulates S phase progression warrants further investigation.

The inability to remove S-phase-specific pADPr eventually led to pADPr throughout cell cycle and γH2AX signaling (*Figure 4C–D*, *Figure 4—figure supplement 1D–F*), indicating that it is essential to resolve S-phase-specific pADPr to maintain cell viability. In agreement with the literature (*Hanzlikova et al., 2018*; *Chen and Yu, 2019*), we observed S-phase-specific pADPr in HeLa cells following PARGi treatment, although such S-phase-specific pADPr was not detected in HEK293A cells following PARGi treatment. The lower level of PARG expression may be the reason for this difference between HEK293A and HeLa cells (*Figure 4—figure supplement 1E*). However, significant amounts of PARG activity still existed in HeLa cells, since these cells did not show noticeable PARGi sensitivity unless we knockout or knockdown PARG.

Cell viability–based genome-wide CRISPR/Cas9 screening unbiasedly uncovers sensitivity and resistance genes under certain conditions (*Zhang et al., 2022*), while FACS-based CRISPR/Cas9 screening using reporter cell lines or antibodies recognizing specific signaling molecules can reveal key regulators of signaling pathways (*Wang et al., 2022*). In this study, we performed both types of

screens, which revealed key pADPr regulators, including BER pathway proteins (POLB, XRCC1, LIG3, and CHD1L), DNA replication/Okazaki fragment maturation proteins (RFCs, FEN1, and LIG1), and ARH3 (*Figure 5B*, *Figure 5—figure supplement 1A*). In addition, consistent with our hypothesis that pADPr accumulation leads to cell death, we showed that loss of genes involved in Okazaki fragment maturation or BER resulted in PARGi sensitivity (*Figure 5C–G*). All of these results further validated our working model that S-phase-specific pADPr likely originated from unligated Okazaki fragments, which if not removed would eventually lead to DNA damage and cell death.

During DNA replication, Okazaki fragments are normally ligated by LIG1. However, some unligated Okazaki fragments may activate PARP1 and recruit XRCC1/LIG3 (*Hanzlikova et al., 2018*), which agrees with the results of early studies indicating that both LIG1 and LIG3 are required for Okazaki fragment maturation (*Arakawa and Iliakis, 2015*). Indeed, we showed that LIG1 depletion, which would result in more DNA nicks in S phase cells, led to the enhancement of S phase PARylation (*Figure 5—figure supplement 1E*). As for LIG3, it can substitute LIG1 function for the ligation of Okazaki fragments (*Arakawa and Iliakis, 2015*; *Paul et al., 2013*). However, it also has a critical and essential function in mitochondria but is separated for its role in XRCC1-dependent SSBR (*Simsek et al., 2011*). We observed minor S phase pADPr due to LIG3 knockdown with PARGi treatment (*Figure 5—figure supplement 1E*), which is likely due to insufficient KD of LIG3 in these experiments. Unlike LIG1 KO and LIG3 KD cells, XRCC1 KO and POLB KO cells increase PARylation in a cell cycle–independent manner (*Figure 5—figure supplement 1E*), which agrees with the finding that POLB and XRCC1 are required for BER/SSB repair, but not specifically for Okazaki fragment ligation/maturation. Our results suggest that PARylation and dePARylation are mainly involved in two cellular processes, that is Okazaki fragment maturation and BER/SSB repair, since both processes have nicked DNA as intermediates, which are likely the physiological substrates that recruit and activate PARP1/2.

Interestingly, two well-known PARylation regulators, ARH3 and HPF1, showed synthetic lethality with PARGi in PARG KO cells (*Figure 5C*). Given that ARH3 and PARG are the primary dePARylation enzymes (*Prokhorova et al., 2021*), it should be predictable that ARH3 was listed as the top hit with PARGi in PARG KO cells in both pADPr signal screen and synthetic lethality screen (*Figure 5B–C*). As for HPF1, a previous report showed that the loss of HPF1 will release the activity of PARP1 to PARylation in acidic resides *Gibbs-Seymour et al., 2016*; moreover, a recent study shows that HPF1 and nucleosomes mediate a dramatic switch in the activity of PARP1 from polymerase to hydrolase *Rudolph et al., 2021*. More important, HPF1 promotes the ligation of the Okazaki fragment by LIG3-XRCC1 as a backup pathway (*Kumamoto et al., 2021*), while the unligated Okazaki fragments were the source of the S phase pADPr signal in this study.

More importantly, our data indicate that PARG expression is a potential biomarker for PARGi sensitivity. We used the available database and cell lines to validate that cells with low PARG expression are sensitive to PARGi. In addition, we examined PARG expression by the IHC assay in commercial TMA samples and uncovered a fraction of breast and ovarian cancers with no detectable or low PARG expression (*Figure 6*). We also examined the sensitivity of HR-deficient cells to PARGi. Although HR deficiency sensitizes cells to PARGi, our results indicate that PARG loss or low expression is still the main driver of PARGi sensitivity in these cells (*Figure 6—figure supplements 1–2*).

Mechanically, we showed that the dePARylation activity of PARG is indispensable for cell survival (*Figure 7*, *Figure 7—figure supplement 2*). The residual PARG dePARylation activity observed in PARG KO cells likely supports cell growth, which can be further inhibited by PARGi (*Figure 7—figure supplement 2*). Although a dose-dependent inhibition of PARG activity by PARGi was also noted in wild-type cells, significant fraction of PARG activity remained even in the presence of highest concentration of PARGi used in this study, which is consistent with our results that HEK293A cells are insensitive to PARGi. More importantly, these data further indicate that PARG expression/activity is a potential biomarker for PARGi sensitivity.

Taken together, the dramatic S-phase-specific pADPr signaling detected under our experimental conditions provides us a rare opportunity to gain a better understanding of PARG function and pADPr signaling in normal proliferating cells without any exogenous DNA damage. In addition, our systematic CRISPR/Cas9 screens uncover key regulators of pADPr signaling that also contribute to PARGi sensitivity. Moreover, our finding that PARG expression may a potential biomarker of PARGi sensitivity will allow the further development of these inhibitors for the treatment of tumors with PARG loss and therefore offer a new targeted strategy for cancer therapy.

## Materials and methods

### Cell culture

HEK293A, HeLa, and OVCAR3 cells were purchased from the American Type Culture Collection (ATCC); HEK293A and HeLa cells were cultured in DMEM with 10% fetal bovine serum, while OVCAR3 cells were cultured in RPMI with 10% fetal bovine serum. RPE1-hTERT FLAG-Cas9 TP53−/− BRCA1−/− cells were a gift that was kindly provided by Dr. Daniel Durocher (University of Toronto) and were cultured in DMEM with 10% fetal bovine serum. RMUGS was obtained from the Japanese Collection of Research Bioresources Cell Bank and cultured in Ham's F12 medium with 10% fetal bovine serum. Knockout cells generated in 293 A and HeLa cells were created with pLentiCRISPRv2 (Addgene, #52961) containing indicated gRNAs (*Figure 7—figure supplement 1*), as described previously (*Wang et al., 2021*). All knockout cells were validated by western blotting and DNA sequencing (*Figure 7—figure supplement 1*). The 293A-derived PARP1/2 DKO cells were the same as those in the previous study (*Wang et al., 2021*). All cell lines were free of mycoplasma contamination.

### Chemical reagents and antibodies

The NAD/NADH Assay Kit II (colorimetric) (ab221821) was obtained from Abcam. The CellTiter-Glo Luminescent Cell Viability Assay (G7573) was purchased from Promega. Nicotinamide (NAM, S1899), β-nicotinamide mononucleotide (NMN, S5259), olaparib (S1060), and FK866 (S2799) were purchased from SelleckChem. PDD 00017272 (HY-133531), PDD 00017238 (HY-133530), PDD 00031705 (HY-135846), and PDD 00017273 (HY-108360) were obtained from MedChemExpress. Methyl methanesulfonate (MMS, 129925), thymidine (T9250), emetine (E2375), and crystal violet solution (HT90132) were from Sigma. The propidium iodide (PI, P3566), GATEWAY cloning system (11789100, 11791100) and FxCycle Violet Stain (F10347) were obtained from ThermoFisher. The QuikChange II Site-Directed Mutagenesis Kit (200523) was from Agilent. The CometAssay Single Cell Gel Electrophoresis Assay (4250–050 K) was obtained from Trevigen.

Antibodies against PARP1 (9532 S), PARG (66564 S, which mainly recognizes full-length isoform 1, but also weakly recognizes isoforms 2 and 3 (data not shown)), and XRCC1 (2735 S) were purchased from Cell Signaling Technology. Antibodies against C-terminus of PARG were purchased from Santa Cruz Biotechnology (SC-398563) and Novus Biologicals (NBP2-55661). Antibodies against histone H3 (ab18521), γH2AX (ab2893), and POLB (ab26343) were from Abcam. Antibodies against β-actin (A5316), tubulin (T6199), and ARH3 (HPA027104) were from Sigma. The antibodies anti-pADPr (10 H) (sc-56198), LIG1 (sc-56087), and BRCA1 (sc-6954) were from Santa Cruz Biotechnology. Anti-PARP2 antibody (39743) was from Creative Motif, anti-LIG3 antibody (GTX70143) was from GeneTex, and anti-53BP1 antibody (NB100-304H) was from Novus Biologicals. Anti-γH2AX (05–6361) and PAR (10 H) (AM80) antibodies were obtained from Millipore. Alexa Fluor Plus 488 secondary antibody goat anti-mouse (A327230) was from Thermo Fisher.

### Plasmid constructs

PARP1 (HsCD00043719) and PARG (HsCD00859023) cDNAs, purchased from DNASU, were subjected to mutagenesis using the QuikChange II Site-Directed Mutagenesis Kit. WT and mutant constructs were subsequently cloned into the vector pLenti CMV Neo DEST (705-1) (Addgene, #17392) by the GATEWAY cloning system. shRNA constructs (pGIPZ-based vector) targeting PARG (Clone ID: V2LHS_11965, V2LHS_250448, V3LHS_379835, and V3LHS_379832) and targeting POLB (Clone ID: V2LHS_170201 and V2LHS_222459) were obtained from Horizon Discovery Biosciences. The Toronto KnockOut Library v3 (TKOv3) (90294) was from Addgene. The DNA Damage Response MKOv4 Library (Addgene, #140219) was from our previous study (*Su et al., 2020*).

### Cell viability assays

To measure cell viability, ~2000 cells were seeded into the 96-well plates. Drugs with indicated concentrations (three biological replicates) were added after 24 hr. After 72 hr of treatment, cell numbers were measured by the CellTiter-Glo Luminescent Cell Viability Assay according to the manufacturer's instructions to draw the cell viability curves.

## NAD+ level measurement

The NAD+ relative level was measured following the manufacturer's instructions. In brief, the same number of cells with the indicated treatment were collected and lysed with NAD+ extraction solution (0.5 M perchloric acid) for 30 min on ice. The extraction was neutralized by 0.55 M $K_2CO_3$ and centrifuged. The supernatant was collected and added to a 96-well plate. NAD+ standard and the NAD+ reaction mixture were added to the 96-well plate for the next colorimetric reading at OD 450 nm after a 30 min reaction. For each time point, at least three biological replicates were measured.

## Clonogenic assay

To assess cellular sensitivity to PARGi or other agents, the indicated cells were seeded on 12-well plates and subsequently exposed to DMSO or indicated treatments for 7–14 days. After phosphate-buffered saline (PBS) washes, cells were stained with crystal violet solution. For each condition, three biological replicates were included and the representative results were presented.

## Western blotting analysis

Unless further operation was indicated, such as chromatin and soluble fractionation, cells were washed with PBS and directly lysed on the plate by 2×Laemmli buffer, boiled at 95 °C for 10 min, and separated by sodium dodecyl sulfate polyacrylamide gel electrophoresis, transferred to membranes, and immunoblotted with the indicated antibodies. For each condition, at least two biological replicates were conducted and the representative results were presented.

## Fluorescence-activated single cell sorting analysis (FACS)

The same number of trypsinized cells with the indicated treatments were collected into a 15 ml tube with PBS, which were subsequently fixed by pre-cold ethanol to a final 70% concentration. After centrifugation, cell pellets were permeabilized by PBS with 0.5% Triton X-100 at ambient temperature for 20 min. After being washed with PBS, cell pellets were blocked by PBS with 4% bovine serum albumin at ambient temperature for 1 hr. The cell pellets were incubated at 4 °C overnight with antibodies (pADPr, AM80; γH2AX, 05–6361) at 1:1000 dilution by PBS with 4% BSA. After being washed with PBS three times, cell pellets were incubated with an Alexa Fluor 488 secondary antibody at 1:500 dilution for another 1 hr at ambient temperature. The pellets were then stained with FxCycle Violet Stain (1 µg/ml) or PI (20 µg/ml) with RNase A (10 µg/ml) and resuspended in PBS for the next flow cytometry analysis by a BD C6 flow cytometer (Becton Dickinson) or Attune Flow cytometers (ThermoFisher). FlowJo software (v10.6.1) was used to analyze the acquired data. For each condition, at least three biological replicates were conducted and the representative results were presented.

## Chromatin and soluble fractionation and enrichment for PARylated proteins

Cells with the indicated treatments were collected for chromatin and soluble fractionation with a protocol reported previously (*Murai et al., 2012*; *Zhang et al., 2022*). In brief, cells were lysed in NETN buffer (20 mM Tris-HCl [pH 8.0], 1 mM EDTA, 100 mM NaCl, 0.5% NP-40, and 1 mM DTT) containing proteinase inhibitors for 20 min on ice. After centrifugation, the soluble fraction (i.e. the supernatant) was collected into fresh tubes. The chromatin fractionation (i.e. the pellets) was washed twice with NETN buffer. The soluble and chromatin fraction were both diluted with 2×Laemmli buffer, boiled at 95 °C for 10 min, and subjected to western blotting analysis. Three biological replicates of chromatin and soluble fractionation were conducted and the representative results were presented.

For enrichment of PARylated proteins in chromatin and soluble fractions, cells were lysed for 20 min on ice by NETN buffer containing proteinase inhibitors and PARP/PARGis (10 µM) to avoid PARylation/dePARylation during cell lysis. After fractionation, the chromatin fractionation (i.e. the pellets) was resuspended by NETN buffer containing proteinase inhibitors and PARP/PARGis (10 µM) and subjected to sonication and centrifugation. The chromatin and soluble fractions were incubated with Af1521 macrodomain affinity resin for 4 hr at 4 °C. After being washed three times with NETN buffer, the beads were boiled with 1×Laemmli buffer at 95 °C for 10 min. The elution was analyzed by western blotting with the indicated antibodies. Two biological replicates of enrichment were conducted and the representative results were presented.

## shRNA knockdown

pGIPZ vectors containing scrambled or gene-specific shRNAs were packed into lenti-virus with the packaging vector psPAX2, the envelope vector pMD2.G, and polyethyleneimine; then lenti-virus was used to infect indicated cell lines. Infected cells were selected with puromycin for ~5 days. The pooled cells were validated by western blotting and used for further experiments.

## Cell viability–based and flow cytometry–based CRISPR screens

DDR library screening was conducted as described previously (*Su et al., 2020*). In brief, cells were infected with DDR library virus at a low multiplicity of infection (MOI) (~ 0.3) for 24 hr and then selected with puromycin (2 μg/ml) for 72 hr. Cells were collected as the initial time point T0, and the remaining cells (five million cells in each replicate, each condition containing at least two replicates) were passaged every 3 days for 21 days. Five million cells were collected at both T0 and the final time point at 21 days (T21).

For whole-genome CRISPR gRNA screening, 120 million cells were infected with lentiviruses encoding the TKOv3 library at a low MOI ratio (<0.3) for 24 hr, as conducted previously (*Zhang et al., 2022*). Infected cells were selected with puromycin (2 μg/ml) for 72 hr. For the cell viability–based screens, 20 million cells were collected after selection and marked as the initial time point (T0). The remaining cells (20 million cells in each replicate, with each condition containing at least two replicates) were passaged every 3 days for 21 days and treated with PARGi or DMSO. Twenty million cells were collected in each condition after 21 days and marked as T21. For flow cytometry-based screening, a similar workflow was performed as described previously (*Wang et al., 2022*). At day 5 after selection, 150 million cells in each replicate were treated with either PARGi (10 μM) for 4 hr or DMSO. Cells were subjected to sequential fixation, permeabilization, blocking, antibody incubation, and staining, as described above. After that, ~20 million cells were collected in both the top 25% with the highest signal and the bottom 25% with the lowest signal cell populations on the basis of the pADPr signal, as determined by flow cytometry.

Genomic DNA was extracted from cell pellets by the QIAamp Blood Maxi Kit (Qiagen) and resuspended in Buffer EB (10 mM Tris–HCl [pH 7.5]) after precipitation by ethanol and sodium chloride. DNA was then amplified by PCR with primers harboring Illumina TruSeq adapters with i5 and i7 barcodes, and the resulting libraries were sequenced on an Illumina NextSeq 500 system. The BAGEL algorithm (https://github.com/hart-lab/bagel, copy archived by *hart-lab, 2024a*) was used to calculate essentiality scores. A DrugZ analysis (https://github.com/hart-lab/drugz, copy archived by *hart-lab, 2024b*) was used to calculate the difference between different groups.

## Alkaline comet assay

An alkaline comet assay was performed using a CometAssay kit under the manufacturer's instructions. In brief, equal numbers of cells from different treatments were collected at the same time and resuspended by PBS with a concentration of $1 \times 10^5$ cells/ml. The cell suspension (50 μl) was well-mixed with 500 μl of molten LMAgarose (at 37 °C) and then immediately spread onto CometSlide. After solidification at 4 °C in the dark for 10 min, slides were incubated with lysis solution for 60 min at 4 °C in the dark and then immersed into the alkaline unwinding solution (200 mM NaOH, 1 mM EDTA, pH >13) for 60 min at 4 °C in the dark. After that, slides were placed into an electrophoresis slide tray with the alkaline unwinding solution for 45 min at 23 voltage. Slides were gently washed, twice with ddH$_2$O and once with 70% ethanol. Slides were dried at 37 °C for ~20 min to make all cells in a single plane and stained with SYBR-Gold. Images were captured by a Nikon 90i microscope at ×10 magnification and analyzed by OpenComet (V1.3.1).

## CRISPR/Cas9-mediated stable knockout cells and reconstitution

Cells were transfected with pLentiCRISPRv2 plasmids containing the gRNAs targeting indicated genes (*Figure 7—figure supplement 1*) using polyethyleneimine. After 24 hr, cells were selected with puromycin for another 48 hr and seeded into 96-well plates with 1 cell in each well. After 2 weeks, the single clones were selected from 96-well plates for further validation by western blotting and DNA sequencing (*Figure 7—figure supplement 1*). Each knockout clone was generated with single gRNA.

pLenti CMV Neo DEST expression vectors containing empty or indicated gene constructor were packed into lenti-virus with the packaging vector psPAX2, the envelope vector pMD2.G,

and polyethyleneimine. Cells were infected with the indicated virus and selected with puromycin for ~5 days, after 24 hr. The pooled cells were confirmed with western blotting to validate the expression of genes and used in further experiments.

## The generation of PARG complete/conditional knockout (cKO) cells

Cells were transfected with pLentiCRISPRv2 plasmids containing the gRNAs targeting the indicated sequences (*Figure 7A*) with polyethyleneimine. After selection with puromycin, cells were seeded into 96-well plates with one cell each well in the presence of low concentration of olaparib (100 nM). Each cKO clone was generated with single gRNA. Viable clones were cultured with low concentration of olaparib and subjected to further validation by western blotting and DNA sequencing (*Figure 7— figure supplement 2*). To keep PARG cKO cells viable, validated clones were always cultured with the addition of low concentration of olaparib.

## Double thymidine block

After cells had adhered to tissue culture plates, they were treated with 2 mM thymidine for 16 hr. The medium was removed, and the cells were washed with sterile PBS twice. Fresh medium was added for another 9 hr. After that, cells were incubated with thymidine for an additional 14 hr.

## Human tissue IHC analysis

Human ovary and breast carcinoma tissue microarrays were purchased from US Biomax. The ovary carcinoma tissue (BC11115d) contains 5 cases of clear cell carcinoma, 62 serous carcinoma, 10 mucinous adenocarcinoma, 3 endometrioid adenocarcinoma, 10 lymph node metastasis carcinoma, 10 adjacent normal ovary tissue (missing one case adjacent normal ovary tissue and one case of serous carcinoma, so we have 98 cases in total for analysis). The breast carcinoma tissue (BC081120f) contains 100 cases of invasive carcinoma of no special type, 10 adjacent normal breast tissue (missing two cases of invasive carcinoma, so we have 108 cases in total for analysis). After deparaffinization and rehydration, antigen retrieval was done by Tris-EDTA Buffer (pH 9.0). The samples were treated with methanol with 1% hydrogen peroxide for 30 min to block endogenous peroxidase activity, followed by incubation with 10% normal goat serum to prevent nonspecific staining; then, the samples were incubated with anti-PARG antibody (1:50. Cell signaling Technology, 24489 S) at 4 °C overnight. The samples were incubated with a biotinylated secondary antibody (1:200. Vector Laboratories, PK-6101) for 30 min at ambient temperature and with avidin–biotin peroxidase complex solution (1:100) for additional 30 min at ambient temperature. The slides were incubated with DAB solution and subsequently counterstained with haematoxylin. The slides were scanned on the Vectra Polaris Automated Quantitative Pathology Imaging System (PerkinElmer, Waltham,US) for quantification by Visiopharm platform (Visiopharm, Hoersholm, Denmark). A total score of protein expression was calculated from both the percentage of immunopositive cells and the immunostaining intensity. High and low protein expressions were defined using the mean score of all normal tissues as a cutoff point.

## DePARylation activity of PARG in whole cell lysates

The dePARylation activity of PARG was measured in the PARP1 Histone H4 Activity Assay (#K611, Tulip BioLabs). Briefly, wells of the plate were incubated with activated PARP1 mixtures to generate PARylated Histone H4 or PARP1 mixtures without $NAD^+$, which were set as the Blank, according to the manufacturer instruction. After that, the plate was washed with PBS three times. Equal number of cells prepared from different cell lines were lysed by equal amount of NETN buffer in the presence of 10 μM olaparib, to inhibit the endogenous PARP1 activity, on ice for 30 min. After centrifugation at 14,300 x *g* at 4 °C for 10 min, supernatant was kept, and the protein concentration was determined by BCA assay. Wells of the plate were either incubated with equal amount of whole cell lysates in NETN buffer with 10 μM olaparib or just NETN buffer with 10 μM olaparib, set as the Control, overnight at room temperature. The Blank was also incubated with equal NETN buffer with 10 μM olaparib. After that, the plate was washed with PBS three times. The remnant pADPr were detected by anti-pADPr antibodies and quantified by TMB substrate with the $OD_{450}$ read, according to the manufacturer instruction. The percentage of pADPr hydrolysis was calculated by normalization with the Blank and the Control.

## Acknowledgements

We thank all of the members of the Chen laboratory for their help and constructive discussions. We also thank Ann Sutton from the Department of Scientific Publications at The University of Texas MD Anderson Cancer Center for editing the manuscript. This work was supported in part by the Pamela and Wayne Garrison Distinguished Chair in Cancer Research to JC. JC also received support from the Cancer Prevention and Research Institute of Texas (CPRIT) (RP160667 and RP180813) and National Institutes of Health (P01 CA193124, R01 CA210929, R01 CA216911, and R01 CA216437). The human tissue IHC analysis was performed in the Flow Cytometry & Cellular Imaging Core Facility, which is supported in part by the National Institutes of Health through MD Anderson's Cancer Center Support Grant CA016672.

## Additional information

### Funding

| Funder | Grant reference number | Author |
|---|---|---|
| Cancer Prevention and Research Institute of Texas | RP160667 | Junjie Chen |
| National Cancer Institute | P01 CA193124 | Junjie Chen |
| National Cancer Institute | R01 CA210929 | Junjie Chen |
| National Cancer Institute | R01 CA216911 | Junjie Chen |
| National Cancer Institute | R01 CA216437 | Junjie Chen |
| Cancer Prevention and Research Institute of Texas | RP180813 | Junjie Chen |

The funders had no role in study design, data collection and interpretation, or the decision to submit the work for publication.

### Author contributions

Litong Nie, Conceptualization, Data curation, Formal analysis, Validation, Investigation, Visualization, Writing - original draft, Writing - review and editing; Chao Wang, Data curation, Formal analysis, Validation, Investigation; Min Huang, Xiaoguang Liu, Xu Feng, Siting Li, Qinglei Hang, Hongqi Teng, Resources, Investigation; Mengfan Tang, Resources; Xi Shen, Investigation; Li Ma, Boyi Gan, Supervision, Writing - review and editing; Junjie Chen, Conceptualization, Supervision, Funding acquisition, Writing - original draft, Project administration, Writing - review and editing

### Author ORCIDs

Litong Nie ⬛ http://orcid.org/0000-0002-0326-4517
Li Ma ⬛ http://orcid.org/0000-0001-9965-989X
Junjie Chen ⬛ http://orcid.org/0000-0002-1493-2189

Reviewer #2 (Public Review): https://doi.org/10.7554/eLife.89303.4.sa1
Reviewer #3 (Public Review): https://doi.org/10.7554/eLife.89303.4.sa2
Author response https://doi.org/10.7554/eLife.89303.4.sa3

## Additional files

### Supplementary files

• Supplementary file 1. The CRISPR-Cas9 screening results. (a) NormZ score of DDR library screen conducted with HEK293A cells treated with PARGi. (b) NormZ score of FACS-based TKOv3 library screen conducted with HEK293A cells treated with PARGi. (c) NormZ score of FACS-based TKOv3 library screen conducted with PARG KO cells. (d) NormZ score of FACS-based TKOv3 library screen conducted with PARG KO cells treated with PARGi. (e) NormZ score

of cell viability–based TKOv3 library screen conducted with HEK293A cells treated with PARGi. (f) NormZ score of cell viability–based TKOv3 library screen conducted with PARG KO cells treated with PARGi.

• MDAR checklist

### Data availability

All data needed to evaluate the conclusions herein are presented in the article or Supplemental Information. All of the knockout and complete knockout cell lines generated in this study are validated by DNA sequencing (*Figure 7—figure supplements 1 and 2B*), which can be requested with the completion of a material transfer agreement. The inducible BRCA1 depletion cells with the use of auxin-inducible degron (mAID) tag cell lines can be provided by J.C. pending scientific review and completion of a material transfer agreement. Requests for these cell lines should be submitted to J.C. via e-mail at JChen8@mdanderson.org.

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
