## [Editor Report · eLife assessment]

The demonstration that the PARG dePARylation enzyme is required in S phase to remove polyADP-ribose (PAR) protein adducts that are generated in response to the presence of unligated Okazaki fragments is potentially **valuable**, but the evidence is **incomplete**, and identification of relevant PARylated PARG substrates in S-phase is needed to understand the role of PARP1-mediated PARylation and PARG-catalyzed dePARylation in S-phase progression.

---

## [Referee Report · Reviewer #2 (Public Review)]

Summary:

In this manuscript Nie et al investigate the effect of PARG KO and PARG inhibition (PARGi) on pADPR, DNA damage, cell viability and synthetic lethal interactions in HEK293A and Hela cells. Surprisingly, the authors report that PARG KO cells are sensitive to PARGi and show higher pADPR levels than PARG KO cells, which is abrogated upon deletion or inhibition of PARP1/PARP2. The authors explain the sensitivity of PARG KO to PARGi through incomplete PARG depletion and demonstrate complete loss of PARG activity when incomplete PARG KO cells are transfected with additional gRNAs in the presence of PARPi. Furthermore, the authors show that the sensitivity of PARG KO cells to PARGi is not caused by NAD depletion but by S-phase accumulation of pADPR on chromatin coming from unligated Okazaki fragments, which are recognized and bound by PARP1. Consistently, PARG KO or PARG inhibition show synthetic lethality with Pol beta, which is required for Okazaki fragment maturation. PARG expression levels in ovarian cancer cell lines correlate negatively with their sensitivity to PARGi.

---

## [Referee Report · Reviewer #3 (Public Review)]

These studies reveal an S-phase requirement for the PARG dePARylation enzyme in removing ADP-ribosylation from PAR-modified proteins whose PARylation is promoted by the presence of unligated Okazaki fragments. The excessive protein ADP-ribosylation observed in S-phase of PARG-depleted human cells leads to trapping of the PARP1 ADP-ribosylation enzyme on chromatin. The findings would be strengthened by identification of the relevant ADP-ribosylation substrates of PARG whose dePARylation is needed for progression through S-phase.

Comments on revised version:

In the revised version the authors have addressed some of the reviewers' concerns, but, despite the new explanatory paragraph on page 16, the paper remains confusing because as shown in Figure 7 at the end of the Results the PARG KO 293A cells that were analyzed at the beginning of the Results are not true PARG knockouts. The authors stated that they did not rewrite the Results because they wanted to describe the experiments in the order in which they were carried out, but there is no imperative for the experiments to be described in the order in which they were done, and it would be much easier for the uninitiated reader to appreciate the significance of these studies if the true PARG KO cell data were presented at the beginning, as all three of the original reviewers proposed.

While the authors have to some extent clarified the nature of the PARG KO alleles, they have not been able to identify the source of the residual PARG activity in the PARG KO cells, in part because different commercial PARG antibodies give different and conflicting immunoblotting results. Additional sequence characterization of PARG mRNAs expressed in the PARG cKO cells, and also in-depth proteomic analysis of the different PARG bands could provide further insight into the origins and molecular identities of the various PARG proteins expressed from the different KO PARG alleles, and determine which of them might retain catalytic activity.

The authors have made no progress in identifying which are the key PARG substrates required for S phase progression, although they suggest that PARP1 itself may be an important target.

---

## [Author Response]

The following is the authors’ response to the previous reviews.

**Reviewer #2 (Recommendations For The Authors):**
I would like to thank the authors for their comments. However, my request for additional experiments to consolidate this manuscript and text changes have not been addressed (point 1 and point 2), which I believe are essential for completion of this manuscript.

The reviewer raised the question about the relevant substrates of PARG in S-phase cells (point 1). As we explained in our previous response, the most important substrate of PARG is PARP1, since we observed increased chromatin-associated PARP1 and PARylated PARP1 in cells with PARG depletion. Moreover, PARP1 or PARP1/2 depletion rescued cell lethality caused by PARG depletion. These data strongly suggest that PARP1 is the major substrate of PARG in S phase cells. Of course, PARG may have additional substrates. In the future, we will perform proteomics experiments as suggested by this reviewer to identify additional PARG substrates, which may reveal new roles of PARG in S phase progression.

The reviewer also suggested us to re-organize our manuscript (point 2). However, we prefer to keep the manuscript as it is, since this is how the project evolved. The other reason we would like to share with the readers is the challenge to validate KO cells. This is an important lesson we learned from this study. We hope that this will raise the awareness of hypomorphic mutant cells we often use to draw conclusions about gene functions and/or genetic interactions. We understand that the current flow of our manuscript may bring some confusion. To avoid it, we included additional explanations at the beginning of this manuscript to draw attention to the readers that our initial KO cells may not be complete PARG KO cells, i.e. they may have residual PARG activity. We also included additional discussion of this important point in the Discussion section.

Moreover, WB analysis of PARG KO clones is inconclusive, as the additional prominent band at 50 kDa could be a degradation product. The authors should check PARG levels are localization by IF, which allows detection of intact proteins and their cellular localizations, since the shorter isoform should be localized in the cytosol. WB with PARG isoforms is missing important information regarding Mw of the PARG constructs and Mw labels of western blots, which makes is difficult to evaluate this data and compare to KO. Ideally, KO and PARG isoform samples should be all on one gel for proper comparison with different antibodies.

We appreciate the concerns raised by this reviewer. We agree that the additional prominent band at 50kDa could be a degradation product. As we explained in our previous response, despite using several PARG antibodies, we could not draw a clear conclusion which functional isoforms or truncated forms were expressed in our PARG KO cells.

Immunostaining experiments may not be more conclusive, since IF experiments rely on the same antibodies for recognizing endogenous PARG. Additionally, even a protein mainly localizes in the cytosol, we cannot exclude the possibility that a small fraction of this protein may localize in nuclei and have nuclear functions.

Instead, as we presented in our manuscript, we used a biochemical assay to measure PARG activity in cell lysate and showed that our initial PARG KO cells still have residual PARG activity. However, we could not detect any PARG activity in our complete/conditional PARG KO cells (cKO cells; these cells can only survive in the presence of PARP inhibitor). These data strongly suggest that PARG is essential for cell survival.